# Neuroprotective effects of hepatoma-derived growth factor in models of Huntington's disease

Kerstin Voelkl[1,2,*] ⓘ, Sara Gutiérrez-Ángel[1,2,*] ⓘ, Sophie Keeling[1,2] ⓘ, Seda Koyuncu[3], Miguel da Silva Padilha[1,2,9], Dennis Feigenbutz[1,2], Thomas Arzberger[4,5,6], David Vilchez[3,7,8] ⓘ, Rüdiger Klein[1] ⓘ, Irina Dudanova[1,2,3,9] ⓘ

Huntington's disease (HD) is a movement disorder caused by a mutation in the Huntingtin gene that leads to severe neurodegeneration. Molecular mechanisms of HD are not sufficiently understood, and no cure is currently available. Here, we demonstrate neuroprotective effects of hepatoma-derived growth factor (HDGF) in cellular and mouse HD models. We show that HD-vulnerable neurons in the striatum and cortex express lower levels of HDGF than resistant ones. Moreover, lack of endogenous HDGF exacerbated motor impairments and reduced the life span of R6/2 Huntington's disease mice. AAV-mediated delivery of HDGF into the brain reduced mutant Huntingtin inclusion load, but had no significant effect on motor behavior or life span. Interestingly, both nuclear and cytoplasmic versions of HDGF were efficient in rescuing mutant Huntingtin toxicity in cellular HD models. Moreover, extracellular application of recombinant HDGF improved viability of mutant Huntingtin-expressing primary neurons and reduced mutant Huntingtin aggregation in neural progenitor cells differentiated from human patient-derived induced pluripotent stem cells. Our findings provide new insights into the pathomechanisms of HD and demonstrate neuroprotective potential of HDGF in neurodegeneration.

## Introduction

Huntington's disease (HD) is a fatal hereditary neurodegenerative disorder that manifests with motor, psychiatric, and cognitive symptoms (Tabrizi et al, 2020). It is caused by a CAG repeat expansion in exon 1 of the Huntingtin gene (The Huntington's Disease Collaborative Research Group, 1993), resulting in translation of the mutant Huntingtin (mHTT) protein with an elongated polyglutamine (polyQ) tract. Neuropathologically, HD is characterized by formation of intranuclear mHTT inclusion bodies (IBs) and by severe neurodegeneration, especially in the striatum and neocortex (DiFiglia et al, 1997). Striatal medium spiny neurons (MSNs) and cortical pyramidal neurons (principal cells [PCs]) belong to the most vulnerable cell types (Vonsattel & DiFiglia, 1998; Waldvogel et al, 2015). Although it is clear that mHTT causes neuronal damage by impairing multiple cellular processes, the exact pathological mechanisms of HD are not yet fully understood (Labbadia & Morimoto, 2013; Saudou & Humbert, 2016; Tabrizi et al, 2020). Moreover, despite promising recent advances in mHTT-lowering therapies (Tabrizi et al, 2019), the clinical trials have so far been unsuccessful, and there is an urgent need for efficient treatments targeting key pathological alterations in HD.

Hepatoma-derived growth factor (HDGF) is a broadly expressed growth factor with neurotrophic activity (Nakamura et al, 1994; Zhou et al, 2004; Marubuchi et al, 2006). HDGF typically localizes in the nucleus, where it can bind DNA and regulate transcription, but it can also be secreted and act in an autocrine or paracrine manner (Everett et al, 2001; Zhou et al, 2004; Yang & Everett, 2007, 2009). The ability of HDGF to prevent neuronal cell death and provide neuroprotection has been demonstrated in nerve lesion models (Marubuchi et al, 2006; Hollander et al, 2012). Of note, altered HDGF expression was reported in a mouse model of motor neuron degeneration and in the brain of human Alzheimer's disease patients (Marubuchi et al, 2006; Bai et al, 2021). However, the disease-modifying potential of HDGF in the context of neurodegenerative disorders including HD has not yet been explored. Moreover, the molecular mechanism of neuroprotection provided by HDGF is poorly understood. Addressing this issue is important in light of recent promise of other growth factors in optimizing therapeutic success against neurodegeneration (De Lorenzo et al, 2020 *Preprint*; Gantner et al, 2020; Albert et al, 2021; Baloh et al, 2022).

[1]Department of Molecules – Signaling – Development, Max Planck Institute for Biological Intelligence, Martinsried, Germany   [2]Molecular Neurodegeneration Group, Max Planck Institute for Biological Intelligence, Martinsried, Germany   [3]Cologne Excellence Cluster on Cellular Stress Responses in Aging-Associated Diseases, University of Cologne, Cologne, Germany   [4]German Center for Neurodegenerative Diseases, Munich, Germany   [5]Center for Neuropathology and Prion Research, Ludwig-Maximilians University Munich, Munich, Germany   [6]Department of Psychiatry and Psychotherapy, Ludwig-Maximilians University Munich, Munich, Germany   [7]Center for Molecular Medicine Cologne, Faculty of Medicine and University Hospital Cologne, University of Cologne, Cologne, Germany   [8]Institute for Integrated Stress Response Signaling, Faculty of Medicine, University Hospital Cologne, Cologne, Germany   [9]Center for Anatomy, Faculty of Medicine and University Hospital Cologne, University of Cologne, Cologne, Germany

Correspondence: ruediger.klein@bi.mpg.de; irina.dudanova@uk-koeln.de
*Kerstin Voelkl and Sara Gutiérrez-Ángel contributed equally to this work

Here, we show that HDGF ameliorates mHTT-related phenotypes in neuron-like cells, primary neurons, and neural progenitor cells (NPCs) derived from induced pluripotent stem cell (iPSC) cultures of HD patients, whereas HDGF deficiency aggravates disease progression in a mouse model of HD. Our results furthermore suggest that nuclear localization of HDGF is not required for its disease-modifying effects. Altogether, our findings uncover neuroprotective properties of HDGF in the context of HD.

## Results

### HDGF reduces mHTT toxicity in PC12 cells and primary neurons

To test whether HDGF is neuroprotective in the context of HD, we first used an inducible stable neuron-like PC12 cell line with pathologically expanded HTT-exon1-Q74 fused to EGFP and a control PC12 cell line with non-pathogenic HTT-exon1-Q23–EGFP (Fig 1A) (Wyttenbach et al, 2001). Induction of mHTT expression in

mHTT cells leads to cell death (Wyttenbach et al, 2001; Hosp et al, 2017). Remarkably, LDH assay revealed a full rescue of cell viability upon HDGF transfection in mHTT cells, whereas the viability of control cells was not further improved (Fig 1B). These data point to a survival-promoting effect of HDGF that is specific to the context of mHTT-induced toxicity.

We next asked whether HDGF also reduced mHTT toxicity in transfected murine primary cortical neurons. Neuronal cell death was assessed by immunostaining with the apoptosis marker active caspase-3 and by DAPI staining, which reveals nuclear fragmentation (Fig 1C). Expression of pathological HTT-exon1-Q97–mCherry caused a significant reduction in survival of transfected neurons compared with the expression of control HTT-exon1-Q25–mCherry or mCherry alone. Consistent with our findings in PC12 cells, this reduction in neuronal survival was rescued by co-expression of HDGF (Fig 1D). In addition, the abundance of mHTT IBs markedly decreased in HDGF-transfected neurons (Fig 1E). These results demonstrate that HDGF ameliorates mHTT-dependent toxicity in neuron-like cells and in primary neurons.

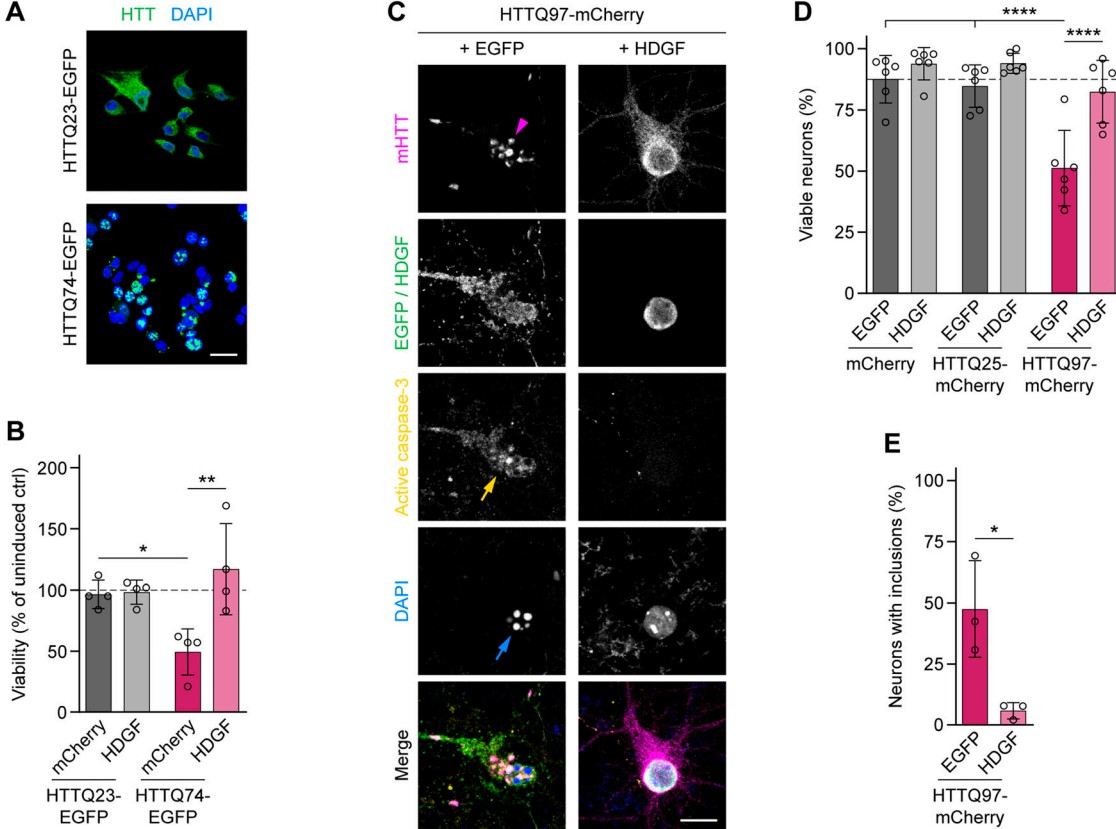

**Figure 1. Rescue of mHTT toxicity by HDGF in PC12 cells and primary neurons.**
**(A)** Images of inducible stable PC12 cell lines expressing control HTT (top) and mHTT (bottom) fused to EGFP. Nuclei were labeled with DAPI. **(B)** Viability of PC12 cells transfected with mCherry or with HDGF was measured 2.5 d after transfection and induction by LDH assay and normalized to uninduced cells. N = 4 independent experiments. Two-way ANOVA with Bonferroni's multiple comparisons test. ANOVA: polyQ, $P = 0.2248$; HDGF, **$P = 0.0088$; polyQ x HDGF, *$P = 0.0118$. **(C)** Cortical neurons transfected with the indicated constructs were fixed at DIV 7 + 2 and stained for cleaved caspase-3. HDGF was detected by immunostaining against Flag-tag; HTT was identified by mCherry fluorescence. Magenta arrowhead points to mHTT IBs, yellow arrow to a caspase-3-positive cell, and blue arrow to nuclear fragmentation revealed by DAPI staining. **(D)** Quantification of the fraction of viable neurons in the indicated conditions. N = 6 independent experiments. Two-way ANOVA with Bonferroni's multiple comparisons test. ANOVA: polyQ, ****$P < 0.0001$; HDGF, ****$P < 0.0001$; polyQ x HDGF, *$P = 0.0107$. **(E)** Fraction of neurons with mHTT IBs. N = 3 independent experiments. Unpaired two-tailed $t$ test. Data information: significant pairwise comparisons are indicated on the graphs. *$P < 0.05$; **$P < 0.01$; ****$P < 0.0001$. Scale bars: (A), 20 $\mu$m; (C), 10 $\mu$m.

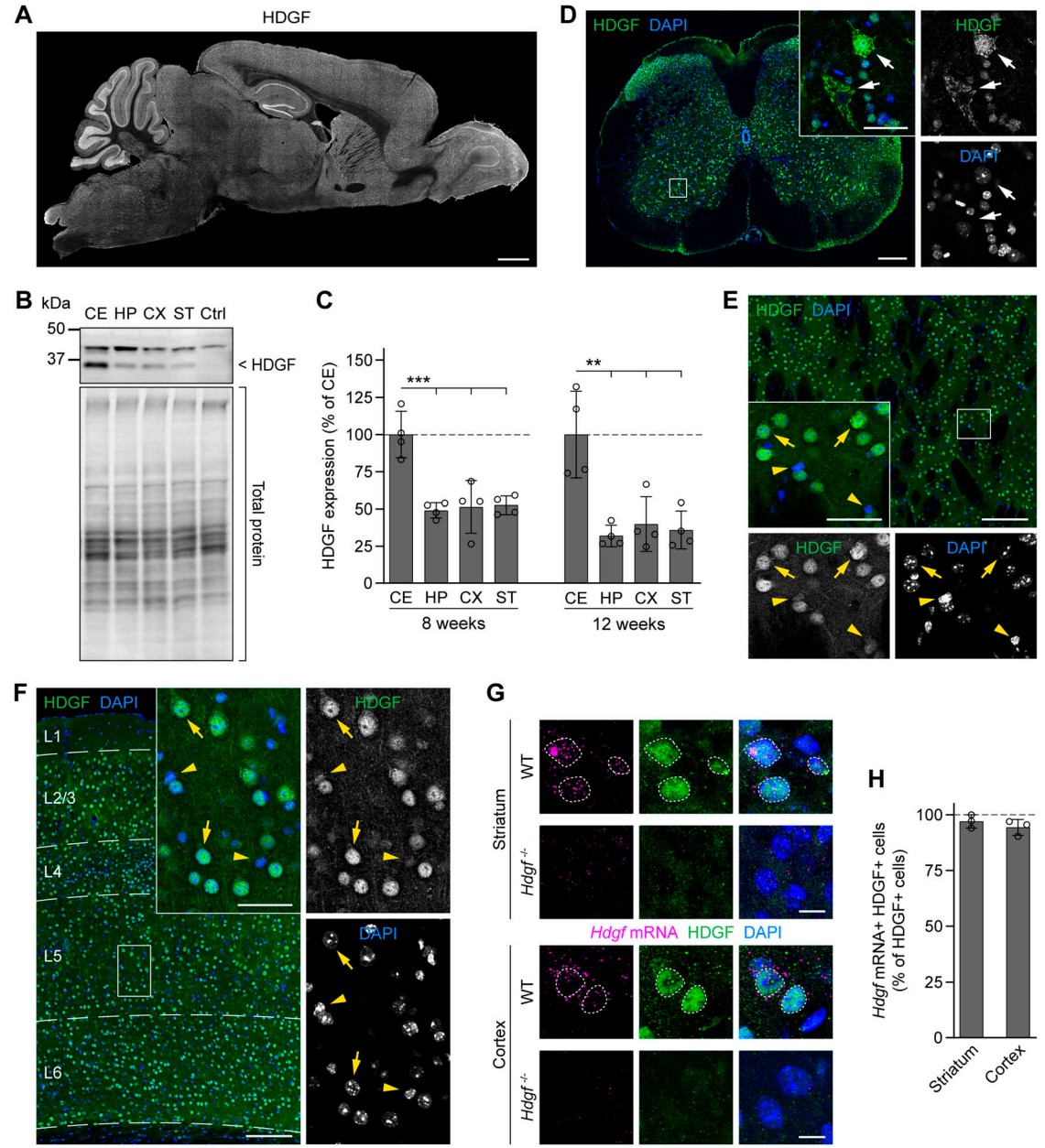

**Figure 2. Overview of HDGF expression in WT mouse central nervous system.**

**(A)** Sagittal brain section from a 12-wk-old C57BL/6 WT mouse immunostained for HDGF. **(B)** Representative Western blot of brain region lysates from 12-wk-old WT mice. HDGF band is indicated on the right; the upper band is unspecific. Total protein detection was used as loading control. Ctrl, cerebellar lysate from $Hdgf^{-/-}$ mice. **(C)** Quantification of HDGF levels in different brain regions of 8-wk-old and 12-wk-old WT mice. Values were normalized to the total protein, followed by normalization to the average value of the region with the highest expression. N = 4 mice per age group. One-way ANOVA with Tukey's multiple comparisons test, per age group. ANOVA: 8 wk, ***P = 0.0002; 12 wk, ***P = 0.0007. CE, cerebellum; HP, hippocampus; CX, cortex; ST, striatum. **(D)** Transverse section of the spinal cord from a 12-wk-old C57BL/6 WT mouse immunostained for HDGF; nuclei were counterstained with DAPI. Dorsal is up. Insets show a higher magnification of the region indicated by the white box. Arrows point to cells in the ventral horn showing cytoplasmic HDGF localization. **(E, F)** Representative images of the dorsal striatum (E) and cerebral cortex (F) immunostained for HDGF from an 8-wk-old C57BL/6 WT mouse. Nuclei were counterstained with DAPI. In (F), cortical layers are indicated on the left and marked with dashed lines. Insets at the bottom (E) and on the right (F) show higher magnifications of the boxed areas. Examples of cells with high levels of HDGF are marked with yellow arrows; examples of cells with low levels of HDGF are marked with yellow arrowheads. Experiments in (A, D, E, F) were performed with N = 4 mice with similar results. **(G)** Images of $Hdgf$ fluorescent in situ hybridization (magenta) combined with HDGF immunostaining (green) in the striatum (top) and cortex (bottom) of 8-wk-old WT and $Hdgf^{-/-}$ mice. Nuclei were counterstained with DAPI. Nuclei positive for the HDGF protein are marked with dashed lines. **(H)** Quantification of the fraction of HDGF-immunopositive cells that also contain $Hdgf$ mRNA. N = 3 mice. Two-tailed one-sample $t$ test. No significant differences were observed. Data information: significant pairwise comparisons are indicated on the graphs. **P < 0.01; ***P < 0.001. Scale bars: (A), 1 mm; (D), 200 μm; (E, F), 150 μm; insets in (D, E, F), 40 μm; (G), 10 μm.

## HDGF expression in the mouse brain

To assess a potential neuroprotective role of HDGF in vivo, we first investigated its expression pattern in the central nervous system of WT mice. Previous studies reported that HDGF is widely expressed during development and in adult tissues, including many regions of the nervous system, where it is found in both neurons and glial cells (Abouzied et al, 2004; Zhou et al, 2004; El-Tahir et al, 2006; Uhlen et al, 2015). However, the expression in different cell types in the brain has not been investigated in detail. Our immunostaining and Western blot experiments in adult C57BL/6 mice revealed a broad expression of HDGF in the brain and spinal cord (Fig 2A–D). Whereas in the forebrain, the subcellular localization of HDGF was mostly nuclear (Fig 2E and F), in the spinal cord, we observed many cells with cytoplasmic expression of HDGF (Fig 2D). HDGF immunoreactivity was specific because the signal was absent in HDGF knockout mice (Fig 2B and G) (Gallitzendoerfer et al, 2008). Interestingly, the levels of HDGF differed between cells (Fig 2E and F). As this protein can be secreted and was proposed to act in a paracrine manner (Oliver & Al-Awqati, 1998; Kishima et al, 2002; Zhou et al, 2004; Thirant et al, 2012; Nusse et al, 2017), we asked whether some of the cells that were positive for HDGF protein might take it up from extracellular space without expressing it endogenously. However, when we combined HDGF staining with fluorescent in situ hybridization, we observed Hdgf mRNA in nearly all the cells that were positive for HDGF protein (Fig 2G and H), suggesting that most cells that contain the HDGF protein also express it endogenously.

To investigate whether levels of HDGF correlated with cellular resistance to HD, we compared HDGF expression in distinct neuronal cell types that show differential susceptibility to degeneration (Fig 3A). In the striatum, we observed higher HDGF expression in HD-resistant ChAT+ cholinergic interneurons (CINs) compared with HD-vulnerable DARPP32+ MSNs (Fig 3B and C). In the cortex, GABAergic interneurons (INs) were genetically labeled by crossing the GAD2-Cre line specific to GABAergic cells (Taniguchi et al, 2011) to the Cre-dependent Ai9 Rosa26-LSL-tdTomato reporter (Madisen et al, 2010), and PCs were detected by neurogranin immunostaining. HDGF expression was significantly higher in the less HD-susceptible cortical INs than in HD-vulnerable PCs (Fig 3B and C). Staining for glial markers furthermore demonstrated that HDGF expression was clearly higher in neurons than in GFAP+ astrocytes, APC+ oligodendrocytes or IBA1+ microglia (Fig S1A–C), suggesting that neurons are the main source of HDGF in the mouse brain. Taken together, these results suggest that higher HDGF expression in the brain might correlate with neuronal resistance to HD.

## HDGF expression is not altered in HD

To test whether HDGF expression is altered in HD, we first evaluated HDGF levels in various brain regions of R6/2 HD mice (Mangiarini et al, 1996) and control littermates by Western blot. No significant differences in HDGF protein quantity were detected in the cerebellum, hippocampus, cortex or striatum at 8 or 12 wk of age (Fig S2A and B). To assess HDGF expression specifically in the neuronal cell types most vulnerable to HD, brain sections of R6/2 and control mice were co-immunostained for HDGF and the MSN marker DARPP32 or the PC marker neurogranin. These experiments revealed no significant changes in HDGF in striatal MSNs or cortical PCs (Fig S2C and D).

To investigate HDGF expression in the context of human HD, we performed immunohistochemistry in postmortem brain tissue from HD patients and age-matched control subjects and analyzed the staining pattern in the primary motor cortex (area M1). HDGF was broadly expressed throughout the cortical layers (Fig S3A). Specificity of the staining was confirmed in control experiments where the first antibody was omitted (Fig S3G). Interestingly, large layer 5 pyramidal neurons, recognized by their large cell body size and the presence of a nucleolus, did not appear to express HDGF, as the observed staining in the cytoplasm was unspecific and likely corresponded to accumulations of lipofuscin (Fig S3C and G). Of note, layer 5 pyramidal cells are among the neurons highly susceptible to HD in the human brain (Waldvogel et al, 2015). Smaller-size neurons displayed diffuse cytoplasmic localization of HDGF (Fig S3B), whereas in glial cells, the staining was more intense and was concentrated in the nucleus

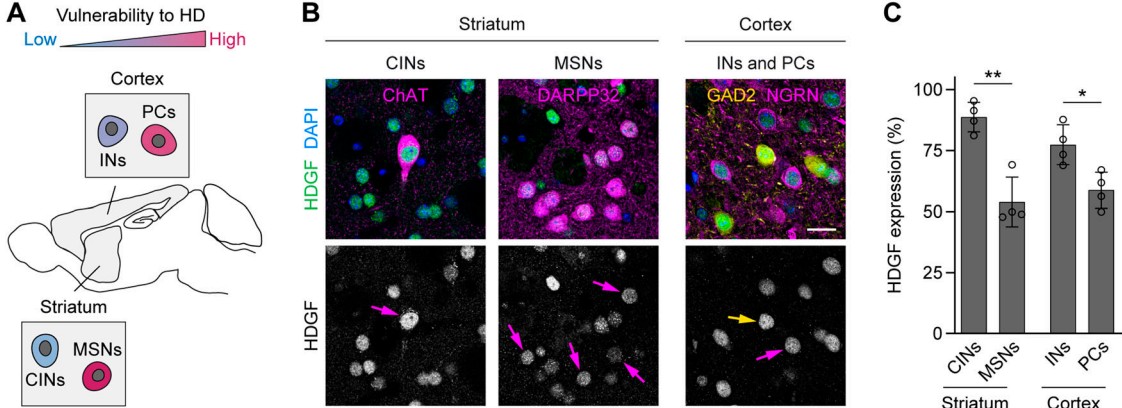

**Figure 3. HDGF expression in HD-vulnerable and resistant neuronal cell types.**
**(A)** Scheme of the mouse brain showing vulnerability of neuronal cell types (insets) to HD. **(B)** Left, brain sections from 8-wk-old C57BL/6 WT mice immunostained against HDGF and indicated striatal neuron markers. Right, sections from 8-wk-old GAD2-Cre; tdTomato mice immunostained against HDGF and neurogranin. Nuclei were counterstained with DAPI. Arrows in the lower row point to the respective cell types. **(C)** Quantification of HDGF staining fluorescence intensity in the indicated cell types. Values were background-subtracted and normalized to the fluorescence intensity of the highest expressing cell in the field of view. N = 4 mice per group. Unpaired two-tailed t test, per brain region. Data information: significant pairwise comparisons are indicated on the graphs. *P < 0.05; **P < 0.01. Scale bar in (B), 20 μm.

(Fig S3B and C). This is in contrast to the expression pattern in mice, where HDGF was localized to the nucleus in all cortical cells and was expressed stronger in neurons than in glia (Figs 2F and S1B). We did not detect any differences in the expression levels or subcellular localization of HDGF between HD and control subjects (Fig S3A–F). Taken together, our findings indicate unchanged levels of HDGF in the brain of HD mice and human patients.

### Deletion of endogenous HDGF exacerbates HD phenotypes in R6/2 mice

The findings that HDGF reduced mHTT toxicity in cultured neurons and its expression levels in the brain appeared to correlate with resistance of neurons to HD raised the hypothesis that endogenous HDGF is neuroprotective. To test this hypothesis, we crossed the R6/ 2 line to $Hdgf^{-/-}$ mice (Fig 4A). R6/2 mice and their WT littermates with and without HDGF deletion underwent a panel of behavioral tests to assess motor skills at 5, 8, and 12 wk of age. In contrast to a previous study that did not detect any motor abnormalities in $Hdgf^{-/-}$ mutants (Gallitzendoerfer et al, 2008), we observed mild hyperactivity of these mice in the open field test, with a statistically significant increase in distance traveled compared with WT at 8 and 12 wk. This hyperactivity was not present in HDGF knockouts crossed to R6/2 mice (Fig 4B). Performance of R6/2 mice on the rotarod was worse than WT controls and was further worsened by HDGF deficiency (Fig 4C). In contrast, we did not observe any significant effect of HDGF ablation on grip strength (Fig 4D). We also did not detect significant changes in mHTT inclusion load, neuronal numbers or overall area of the striatum and cortex (Fig S4A–E). Importantly, HDGF deficiency caused a significant reduction in the life span of

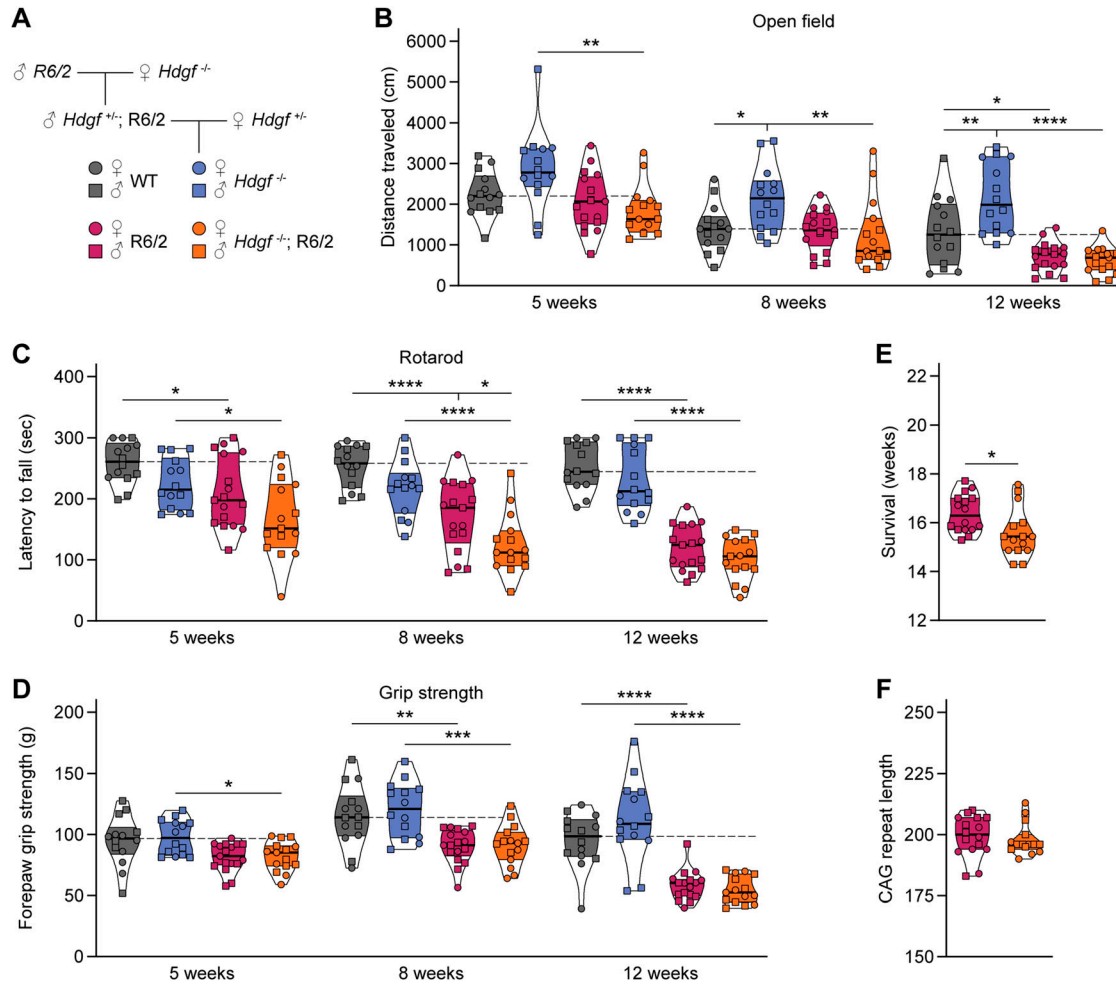

**Figure 4. Genetic ablation of HDGF exacerbates motor defects and shortens life span in R6/2 mice.**
**(A)** Breeding scheme for $Hdgf^{-/-}$; R6/2 mutants. **(B)** Total distance traveled in the open field. Repeated measures three-way ANOVA with Bonferroni's multiple comparisons test per age group. ANOVA: age, ****$P < 0.0001$; polyQ, ****$P < 0.0001$; HDGF, $P = 0.0560$; age x polyQ, ***$P = 0.0008$; age x HDGF, $P = 0.5773$; polyQ x HDGF, **$P = 0.0054$; age x polyQ x HDGF, $P = 0.9482$. **(C)** Latency to fall from the rotarod. Repeated measured three-way ANOVA with Bonferroni's multiple comparisons test per age group. ANOVA: age, ****$P < 0.0001$; polyQ, ****$P < 0.0001$; HDGF, **$P = 0.0017$; age x polyQ, ****$P < 0.0001$; age x HDGF, $P = 0.1624$; polyQ x HDGF, $P = 0.7995$; age x polyQ x HDGF, $P = 0.8124$. **(D)** Forelimb grip strength. Repeated measured three-way ANOVA with Bonferroni's multiple comparisons test per age group. ANOVA: age, ****$P < 0.0001$; polyQ, ****$P < 0.0001$; HDGF, $P = 0.2832$; age x polyQ, ****$P < 0.0001$; age x HDGF, $P = 0.7028$; polyQ x HDGF, $P = 0.2008$; age x polyQ x HDGF, $P = 0.2316$. **(E)** Survival. Unpaired two-tailed $t$ test. **(F)** CAG repeat length. Unpaired two-tailed $t$ test, not significant. N = 14 WT mice, 14 $Hdgf^{-/-}$ mice, 16–17 R6/2 mice and 15 $Hdgf^{-/-}$; R6/2 mice for all analyses in (B, C, D, E, F). Data information: significant pairwise comparisons are indicated on the graphs. *$P < 0.05$; **$P < 0.01$; ***$P < 0.001$; ****$P < 0.0001$.

R6/2 mice by 6 d on average (Fig 4E). CAG repeat sizing confirmed a similar length of the CAG tract in $Hdgf^{+/+}$ and $Hdgf^{-/-}$ cohorts of R6/2 mice (Fig 4F), excluding any effects of CAG expansion size on the phenotype. Taken together, these experiments demonstrate that HDGF deficiency causes behavioral abnormalities in WT mice, and aggravates motor deficits and shortens life span in R6/2 mice.

## Effects of HDGF overexpression in HD mice

To assess whether overexpression of HDGF ameliorates HD phenotypes in vivo, we performed stereotactic injections of AAV8-EYFP-P2A-Flag-HDGF or AAV8-EYFP control virus into the dorsal striatum of 4-wk-old R6/2 mice and WT littermates (Fig S5A). This resulted in a prominent increase in local HDGF levels in the striatum of HDGF-injected mice (Fig S5B and C). At 12 wk of age (8 wk after the AAV injections), the mice were assessed in the open field and rotarod tests. As expected, EYFP-injected R6/2 mice showed markedly impaired locomotion in the open field compared with WT littermates,

with a significant reduction in the distance traveled. This phenotype was not rescued in HDGF-injected R6/2 mice (Fig S5D). In the rotarod, both YFP- and HDGF-injected R6/2 mice were significantly impaired in comparison with the WT groups (Fig S5E). Likewise, the life span of R6/2 mice was not changed upon striatal delivery of HDGF (Fig S5F). Interestingly, histological assessment revealed significantly reduced size of mHTT IBs within the injected area of the striatum (Fig 5A and B). Altogether, these data suggest that local delivery of HDGF to the striatum of juvenile mice leads to a reduction in mHTT inclusion size, but does not cause a major change in neurological phenotypes.

We reasoned that an earlier and broader overexpression of HDGF might be required for modifying HD symptoms in mice. As HD is an inherited disease, gene-expansion carriers can be identified and preventive treatments started at an early age. We therefore evaluated the efficiency of HDGF treatment given to newborn pups. To this end, EYFP-P2A-Flag-HDGF or EYFP control was overexpressed throughout the brain by AAV injections into the lateral

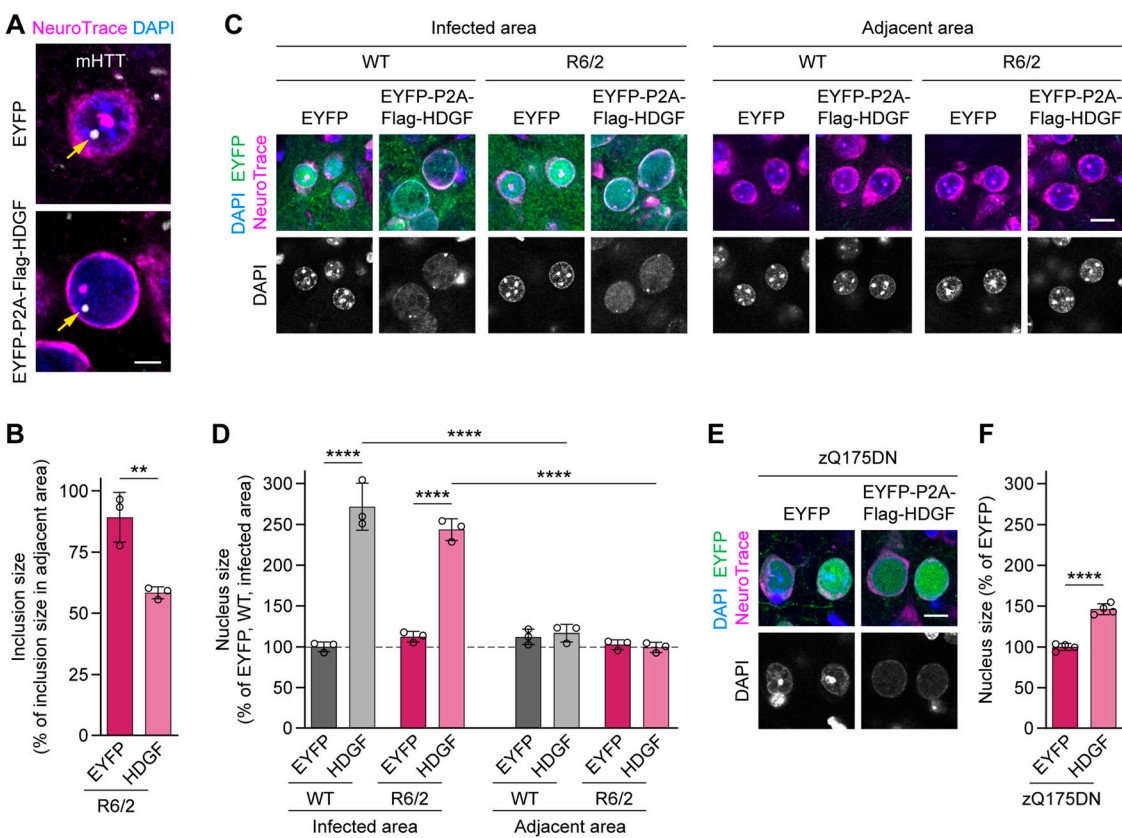

**Figure 5. Overexpression of HDGF reduces the size of mHTT inclusions and causes nuclear expansion.**
**(A)** Representative images of striatal neurons in 12–13-wk-old R6/2 mice striatally injected with EYFP (top) or HDGF (bottom). mHTT was detected with EM48 immunostaining, neurons were identified with NeuroTrace, nuclei were counterstained with DAPI. Arrows point to nuclear mHTT inclusions. **(B)** Quantification of neuronal mHTT IB size within the infected area of EYFP- and HDGF-injected R6/2 mice, normalized to IB size in the striatum outside the infected area. N = 3 mice. Unpaired two-tailed t test. **(C)** Images of striatal neurons from 12–13-wk-old WT and R6/2 mice striatally injected with the indicated constructs, inside (left) and outside (right) the infected area. Neurons were identified with NeuroTrace, nuclei were counterstained with DAPI. Note the increased nuclear size and altered DAPI staining pattern within the area infected with HDGF. **(D)** Quantification of nucleus size. The data were normalized to the average nucleus size in the EYFP-infected area of WT mice. N = 3 mice per group. Three-way ANOVA with Bonferroni's multiple comparisons test. ANOVA: area, ****P < 0.0001; polyQ, P = 0.0621; HDGF, ****P < 0.0001; area x polyQ, P = 0.6099; area x HDGF, ****P < 0.0001; polyQ x HDGF, *P = 0.0376; area x polyQ x HDGF, P = 0.1480. **(E)** Images of cortical neurons from 13-mo-old zQ175DN mice injected with the indicated constructs at P0. Neurons were identified with NeuroTrace, nuclei were counterstained with DAPI. Note the increased nuclear size and altered DAPI-staining pattern with HDGF. **(F)** Quantification of nucleus size. The data were normalized to the average nucleus size in the EYFP-injected mice. N = 4 mice. Unpaired two-tailed t test. Data information: significant pairwise comparisons are indicated on the graphs. **P < 0.01; ****P < 0.0001. Scale bars: (A), 5 μm; (C, E), 10 μm.

ventricle of postnatal day 0 (P0) pups. P0 injections resulted in a broad expression of the exogenous Flag–HDGF protein, with particularly high levels in the hippocampus and cortex (Fig S6A–D). Behavioral assessment at 12 wk did not reveal significant changes in the motor performance in HDGF-injected mice. The slight increase in distance traveled in the open field test did not reach statistical significance (Fig S6E). No improvements were observed in the rotarod and grip strength tests (Fig S6F and G), and the life span of HDGF-injected R6/2 mice also remained unchanged (Fig S6H). Taken together, our findings in R6/2 mice suggest that overexpression of HDGF in the nervous system does not have a significant effect on HD-related neurological phenotypes.

In addition to the early-onset transgenic R6/2 model, which expresses an N-terminal fragment of mHTT, we investigated the consequences of HDGF overexpression in the late-onset zQ175DN knock-in mouse model (Menalled et al, 2012; Southwell et al, 2016) that expresses full-length mHTT from the endogenous murine *Htt* locus and, therefore, more faithfully reproduces human HD. AAV8-EYFP-P2A-Flag-HDGF or AAV8-EYFP were stereotactically injected into the lateral ventricle of heterozygous zQ175DN knock-in mice and WT littermate controls at P0, and motor behavior in the open field, rotarod, and grip strength tests was evaluated at 12 mo of age. HDGF-injected WT mice displayed increased locomotion in the open field and an increase in forepaw grip strength (Fig S7A–C). However, heterozygous zQ175DN mice did not show significant deterioration of motor performance in any of the tests (Fig S7A–C), hence, a putative protective effect of HDGF could not be evaluated in this line.

## Nuclear localization of HDGF is not required for ameliorating mHTT toxicity in neurons

We noticed that HDGF overexpression caused an increase in the nucleus size and a change in the DAPI staining pattern in both R6/2 and zQ175DN mice (Fig 5C–F). Enlarged nuclear size and altered chromatin organization are morphological hallmarks of several types of cancer (Jevtic & Levy, 2014). We therefore asked whether the positive effects of HDGF on mHTT-dependent phenotypes could be separated from its impact on the nuclear size. To this end, we generated a cytosolic version of HDGF by introducing nine amino acid substitutions into the two nuclear localization sequences of HDGF (Kishima et al, 2002), and adding a nuclear export sequence at its C-terminus (cytHDGF, Fig S8A). Immunostaining of transduced primary neurons demonstrated that cytHDGF was excluded from the nucleus and localized only in the cytoplasm (Fig S8B). Exclusion of HDGF from the nucleus completely abolished its effect on the nuclear size (Fig S8C). We then co-expressed wtHDGF and cytHDGF with mHTT in primary neurons. In addition to the HTT-exon1-Q97-mCherry version of mHTT described above (Fig 1C–E), in these experiments we also used HTT-exon1-Q72-His. This construct has a different tag and different polyQ length, and also forms mHTT inclusions predominantly in the nucleus, whereas HTT-exon1-Q97-mCherry inclusions are mostly found in the cytoplasm (Fig 6A). Both versions of mHTT cause comparable neurotoxicity (Fig 6B). Remarkably, both wtHDGF and cytHDGF significantly increased the survival of neurons expressing either of the mHTT constructs,

indicating that the toxicity-modifying effects of HDGF are not dependent on mHTT tag or polyQ length. The degree of rescue was comparable for the two versions of HDGF (Fig 6B). In addition, both HDGF variants decreased the fraction of neurons with HTT-exon1-Q97-mCherry inclusions (Fig 6C). Although the overall fraction of cells bearing HTT-exon1-Q72-His inclusions was not changed, the percentage of cells with large (≥1 μm) inclusions was significantly reduced (Fig 6C and D). These results indicate that nuclear localization of HDGF is not required for mitigating mHTT toxicity in neurons.

To test whether cytHDGF can improve HD phenotypes in vivo, we delivered AAV8-EYFP-P2A-Flag-cytHDGF or AAV8-EYFP control virus into the nervous system of newborn R6/2 and littermate control pups (Fig S9A–D). Consistent with the cell culture results, we observed the cytoplasmic localization of overexpressed cytHDGF in the brain (Fig S9A) and unaltered nuclear morphology of cytHDGF-infected neurons (Fig S9B). The mice were subjected to behavioral tests (open field, rotarod, grip strength) at 12 wk of age. There was no significant improvement in the motor skills of cytHDGF-injected R6/2 mice (Fig S9E–G). Taken together, our experiments show that cytHDGF has a similar impact on mHTT toxicity as wtHDGF, although not causing nuclear expansion.

## Extracellular HDGF mitigates mHTT-dependent phenotypes in primary mouse neurons and NPCs derived from human iPSCs

As HDGF can be secreted and act in a paracrine manner (Nakamura et al, 1994; Oliver & Al-Awqati, 1998; Zhou et al, 2004; Thirant et al, 2012), we investigated the potential of extracellular HDGF to modify HD phenotypes in cell culture. To this end, we produced recombinant HDGF and first tested its ability to rescue the survival of primary neurons in a starving medium. Recombinant brain-derived neurotrophic factor (BDNF) served as positive control. In agreement with a previous report (Zhou et al, 2004), we observed a significant increase in neuronal survival in the presence of extracellular HDGF (Fig 7A), confirming that our recombinant HDGF protein is biologically active. We then added recombinant wtHDGF or cytHDGF to dissociated neuronal cultures transfected with pathological (HTT-exon1-Q97-mCherry) or control (HTT-exon1-Q25-mCherry) HTT constructs and measured cell viability 2 d later. BDNF, which is known to improve survival of mHTT-expressing neurons, was used as positive control. mHTT-expressing neurons treated with either version of recombinant HDGF showed a significant increase in viability (Fig 7B).

As growth factors often exert their effects by activating canonical ERK1/2 and PI3K/AKT intracellular signaling pathways, we monitored the activation of these pathways in WT neurons 10–20 min after application of recombinant HDGF and BDNF. Western blot analysis showed that whereas BDNF clearly increased both p-ERK1/2 and p-AKT levels, HDGF failed to activate any of the two pathways (Fig S10A–C). These results suggest that extracellular HDGF ameliorates mHTT toxicity in primary neurons without activating ERK1/2 or PI3K/AKT signaling.

We then asked whether HDGF can rescue mHTT-induced phenotypes in a human model system. iPSCs from a patient with juvenile HD (HTTQ71) and control iPSCs without pathological HTT

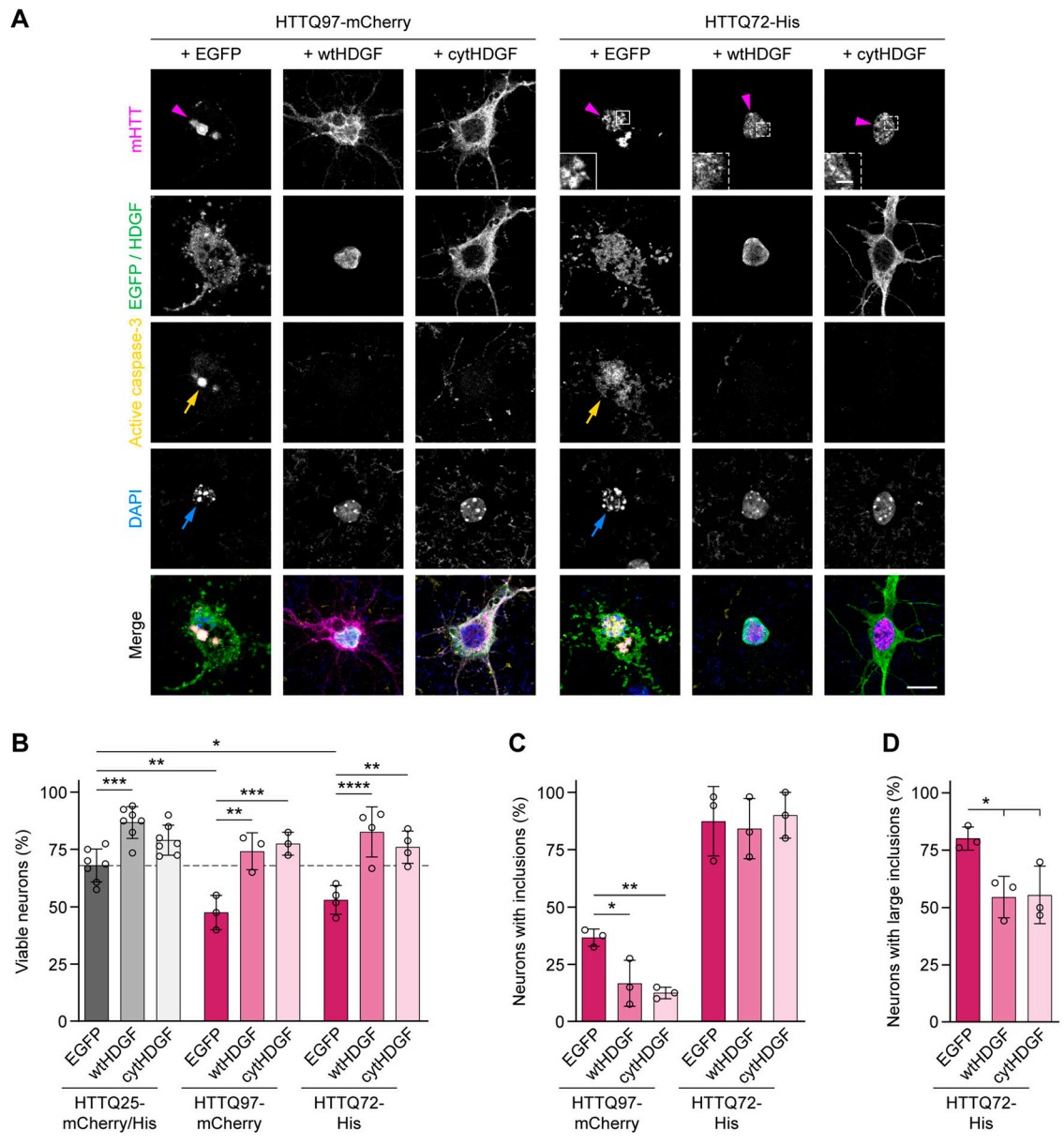

**Figure 6. Nuclear localization of HDGF is not required for ameliorating mHTT toxicity.**
**(A)** Cortical neurons transfected with the indicated constructs were fixed at DIV 7 + 2 and stained for cleaved caspase-3. HDGF was detected by immunostaining against Flag-tag; HTT was identified by mCherry immunofluorescence or by immunostaining against His-tag. Magenta arrowheads point to mHTT IBs, yellow arrows to caspase-3-positive cells, and blue arrows to nuclear fragmentation revealed by DAPI staining. Insets show magnifications of boxed areas. Note the difference between large IBs (solid line box) and small foci (dashed line boxes) formed by HTTQ72-His. **(B)** Quantification of the fraction of viable neurons in the indicated conditions. HTTQ25-mCherry and HTTQ25-His controls did not differ from each other and were pooled for analysis. N = 3–7 independent experiments. Two-way ANOVA with Bonferroni's multiple comparisons test. ANOVA: polyQ, ***$P$ = 0.0007; HDGF, ****$P$ < 0.0001; polyQ x HDGF, $P$ = 0.0818. **(C)** Fraction of neurons with mHTT IBs. N = 3 independent experiments. One-way ANOVA with Tukey's multiple comparisons test per mHTT construct. ANOVA: HTTQ97-mCherry, **$P$ = 0.0077; HTTQ72-His, $P$ = 0.1510. **(D)** Fraction of neurons with large mHTT IBs. N = 3 independent experiments. One-way ANOVA with Tukey's multiple comparisons test. ANOVA: *$P$ = 0.0258. Data information: significant pairwise comparisons are indicated on the graphs. *$P$ < 0.05; **$P$ < 0.01; ***$P$ < 0.001; ****$P$ < 0.0001. Scale bar in (A), 10 $\mu$m; insets, 2 $\mu$m.

expansion (HTTQ33) were differentiated into NPCs using established protocols. Treatment with the proteasome inhibitor MG-132 significantly increased aggregation of mHTT in HTTQ71-NPCs as revealed by filter trap assay (Fig 7C and D) and led to appearance of IBs detectable by immunostaining (Fig S11A and B), in agreement with a previous study (Koyuncu et al, 2018). Application of both recombinant wtHDGF and cytHDGF abolished

the significant increase in mHTT aggregation detected with filter trap (Fig 7C and D). Accordingly, immunostaining showed a decrease in the fraction of cells with mHTT IBs in HTTQ71-NPCs treated with either wtHDGF or cytHDGF (Fig S11A and B). In summary, these results demonstrate that extracellular HDGF rescues mHTT-dependent phenotypes in mouse primary neurons and human NPC cultures.

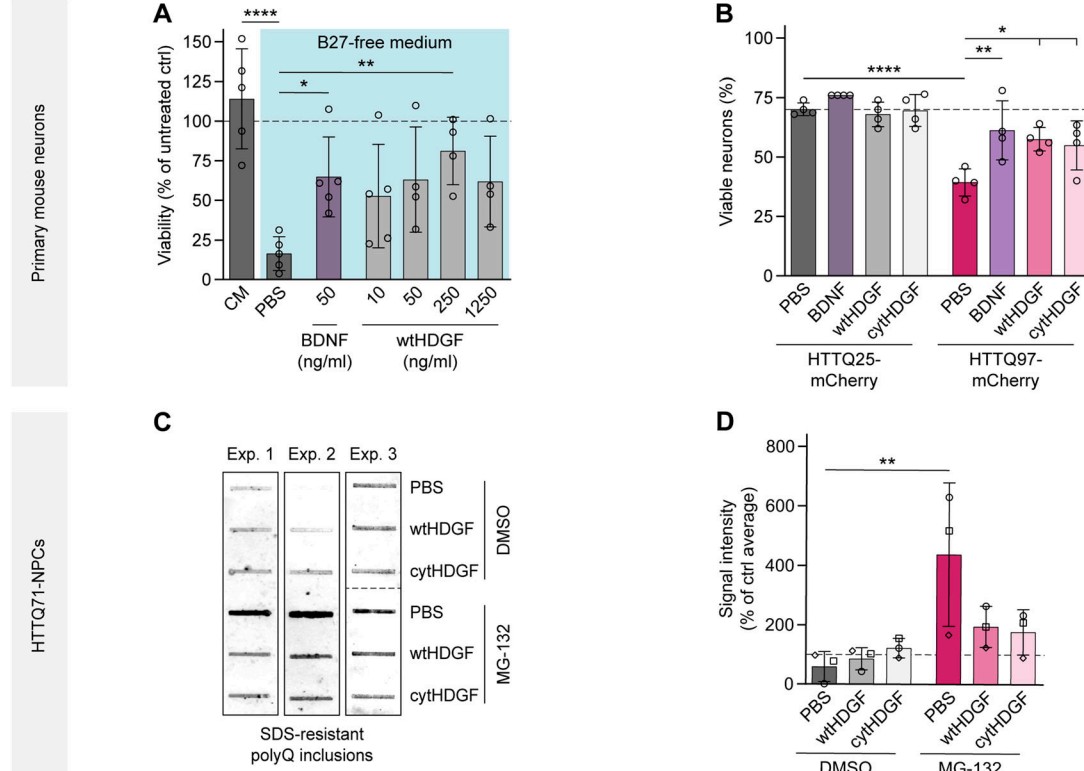

**Figure 7. Extracellular HDGF mitigates mHTT-dependent phenotypes in primary mouse neurons and human NPCs.**
**(A)** Viability of primary mouse cortical neurons was measured 24 h after medium exchange by MTT assay and normalized to untreated cells. Medium was replaced at DIV 7 by complete culture medium (CM) or B27-free culture medium supplemented with PBS, recombinant BDNF or recombinant wtHDGF at the indicated concentrations. N = 4–5 independent experiments. One-way ANOVA with Dunnett's multiple comparisons test. ANOVA: ***P = 0.0007. Significant pairwise comparisons to PBS treated cells are indicated on the graph. **(B)** Cortical neurons transfected with the indicated HTT constructs and treated with PBS, recombinant BDNF (50 ng/ml) or HDGF (250 ng/ml) were fixed at DIV 7 + 2 and stained for cleaved caspase-3. Nuclear fragmentation was revealed by DAPI staining. The fraction of viable neurons was quantified. N = 4 independent experiments. Two-way ANOVA with Bonferroni's multiple comparisons test. ANOVA: polyQ, ****P < 0.0001; treatment, **P = 0.0072; polyQ x treatment, *P = 0.0408. **(C)** Filter trap of polyQ aggregates in HDQ71-NPCs derived from iPSCs. Aggregation was induced 24 h after treatment with PBS or recombinant HDGF (250 ng/ml) by proteasome inhibition with MG-132 for 8 h. Blot on the right was digitally rearranged from horizontal to vertical order indicated by the dashed line. **(D)** Quantification of the filter trap assay. N = 3 independent experiments. Two-way ANOVA with Bonferroni's multiple comparisons test. ANOVA: MG-132, **P = 0.0048; HDGF, P = 0.2175; MG-132 x HDGF, P = 0.0579. Data information: significant pairwise comparisons are indicated on the graphs. *P < 0.05; **P < 0.01; ****P < 0.0001.

## Discussion

Here, we show that HDGF mitigates mHTT toxicity and reduces mHTT aggregation in mouse and human cellular models of HD. Our cell type-specific expression studies in the cortex and striatum of WT mice suggest that HDGF expression levels might inversely correlate with the susceptibility of neuronal cell types to HD. In agreement with this, in the human postmortem cortical tissue, HDGF appeared to be absent in large layer 5 pyramidal neurons that are the main source of the corticostriatal projection and are highly vulnerable to HD (Waldvogel et al, 2015).

We further find that endogenous HDGF is neuroprotective in vivo, because ablation of HDGF exacerbates HD phenotypes in R6/2 mice. This is consistent with the results of a recent in vivo shRNA screen, where *Hdgf* was among the essential genes required for survival of striatal neurons in healthy and HD mice (Wertz et al, 2020). However, overexpression of HDGF was not sufficient to significantly improve neurological disease phenotypes. One potential factor contributing to the modest beneficial effects of HDGF overexpression in vivo might be that its levels were only increased in the brain, but not in the peripheral tissues. As HTT is ubiquitously expressed, its mutation causes pathological changes in multiple tissues beyond the brain including the musculoskeletal and cardiovascular systems, which may play an important role in disease manifestations (van der Burg et al, 2009). Another possible explanation is that the relatively high endogenous expression of HDGF in the nervous system saturates the activation of its receptors and/or signaling pathways, precluding additional benefit from ectopic expression. Alternatively, increased HDGF concentration alone might not be sufficient to rescue HD phenotypes, if the downstream signaling components are limited or impaired because of the HD mutation. Indeed, such a scenario was proposed for BDNF-dependent deficits in HD, where compromised postsynaptic signaling downstream of the BDNF receptor TrkB contributes to the impaired striatal physiology (Plotkin et al, 2014). The putative HDGF receptor and the downstream signaling pathway are currently unclear and remain to be explored. Elucidating this downstream mechanism could reveal new treatment approaches for HD. Moreover, although HDGF might not be sufficient as a therapy on its own, it could still be beneficial in combination with other treatment approaches because of its neurotrophic effects. Thus, viral

delivery of glial cell line-derived neurotrophic factor (GDNF) clearly improved the survival and functional integration of transplanted human stem cells in a rodent model of Parkinson's disease (Gantner et al, 2020).

In our experiments with heterozygous zQ175DN mice, we failed to observe significant motor defects in this line at 12 mo of age (Fig S7). Although some studies described mild motor impairments in heterozygous zQ175 mice (Menalled et al, 2012; Southwell et al, 2016), other studies also reported largely unchanged motor behavior at a similar age (Heikkinen et al, 2012; Zeitler et al, 2019). Our results suggest that behavioral assessments alone might not be sufficient in such a slowly progressing model, and they need to be combined with molecular, electrophysiological, and/or morphological analyses to be more conclusive.

Long-term overexpression of HDGF in the in vivo experiments led to a significant change of nuclear size in both R6/2 and zQ175DN models, an effect that could be circumvented by targeting HDGF to the cytoplasm. Importantly, the cytoplasmic version of HDGF proved equally efficient in ameliorating mHTT neurotoxicity and reducing its aggregation in primary neurons and human iPSC-derived neural progenitors. Our expression studies show that subcellular localization of endogenous HDGF differs between cell types and between mouse and human brains. These findings raise several exciting questions, including how the subcellular localization of HDGF is regulated in different cell types. Moreover, it remains to be determined whether the mechanisms of action of nuclear and cytoplasmic HDGF are distinct or overlapping, how this relates to the different mechanisms of mHTT toxicity in these subcellular compartments (Landles et al, 2020; Blumenstock et al, 2021), and ultimately influences the vulnerability of respective cells to degeneration. It will also be interesting to investigate the therapeutic effects of HDGF in other animal models where the subcellular distribution of the protein is more similar to that in the human brain.

A number of other growth factors have been proposed for HD treatment, including BDNF, ciliary neurotrophic factor, and fibroblast growth factor 9 (Anderson et al, 1996; Emerich et al, 1997; Mittoux et al, 2000; Zuccato & Cattaneo, 2007; Xie et al, 2010; Yusuf et al, 2018). All these growth factors are believed to act by triggering canonical intracellular signaling cascades such as AKT and ERK pathways upon binding to their cell-surface receptors. In contrast, our results suggest that the neuroprotective activity of HDGF is independent of these canonical pathways (Fig S10). One other growth factor, insulin-like growth factor 2, was also proposed to act through a noncanonical mechanism by stimulating the secretion of mHTT aggregates through extracellular vesicles (Garcia-Huerta et al, 2020). In future experiments, it will be important to determine the exact molecular mechanism of HDGF effects in HD models, and to explore its neuroprotective potential in other neurodegenerative disorders beyond HD.

# Materials and Methods

## Plasmids

The following plasmids were used for transfection: pcDNA3.1 mCherry, HTT-exon1-Q25-mCherry-Myc-His, and HTT-exon1-Q97-mCherry-Myc-His (Hipp et al, 2012); pcDNA3.1 HTT-exon1-Q25-His, and HTT-exon1-Q72-His (cloned from pPGK HTT-exon1-Q25, HTT-exon1-Q72) (Jeong et al, 2011); pCI-neo EGFP-HA (cloned from pCI-neo Fluc-EGFP) (Gupta et al, 2011); pCMV6-Entry HDGF-Myc-Flag (RC204148; OriGene); and cytHDGF-Myc-Flag. To generate cytHDGF, 12 point mutations were introduced into the two nuclear localization signals of HDGF by gene synthesis (Kishima et al, 2002), and a nuclear export sequence (SELQNKLEELDLDSYK) (Goedhart et al, 2012) was added C-terminally. For lentiviral expression, EYFP, EYFP-P2A-Flag-HDGF, and EYFP-P2A-Flag-cytHDGF were synthesized and cloned into pFhSynW2 (kindly provided by Dieter Edbauer) (May et al, 2014). psPAX2 and pcDNA3.1 VSV-G (kindly provided by Dieter Edbauer) were used for lentiviral packaging. Adenoviral plasmids were generated at the Viral Vector Production Unit, Universitat Autònoma de Barcelona: pAAV-CAG EYFP, EYFP-P2A-Flag-HDGF, and EYFP-P2A-Flag-cytHDGF. For recombinant protein production, HDGF and cytHDGF with a C-terminal Flag-TEV-His sequence were cloned into pET-17b (kindly provided by Kathrin Lang).

## Mouse lines

All animal experiments were approved by the Government of Upper Bavaria, Germany (permit numbers 55.2-1-54-2532-168-14, 55.2-1-54-2532-19-2015, ROB-55.2-2532.Vet_02-20-05, and ROB-55.2-2532.Vet_02-19-083) and conducted in accordance with the relevant guidelines and regulations. Mice were housed with ad libitum access to food and water in the animal facilities of the Max Planck Institute for Biological Intelligence. Transgenic R6/2 mice (Mangiarini et al, 1996) (#002810; JAX stock) were maintained by breeding hemizygous R6/2 males with the female F1 progeny of a cross between CBA (Janvier Labs) and C57BL/6 (Janvier Labs) mice. zQ175DN (Menalled et al, 2012; Southwell et al, 2016) (#029928; JAX stock), $Hdgf^{-/-}$ (Gallitzendoerfer et al, 2008) (kindly provided by Sørge Kelm), GAD2-Cre (Taniguchi et al, 2011) (#010802; JAX stock), and Ai9 tdTomato (Madisen et al, 2010) (#007909; JAX stock) mice were kept on C57BL/6 background. To generate HDGF-deficient R6/2 mice, hemizygous R6/2 males were crossed to homozygous $Hdgf^{-/-}$ females, followed by mating the F1 offspring R6/2 transgenic males with heterozygous $Hdgf^{+/-}$ females. For genetic labeling of cortical interneurons, heterozygous GAD2-Cre mice were crossed to homozygous tdTomato reporter mice. Immunostainings shown in Figs 2A and D and S2C and D, Western blots shown in Fig S5B and C, and survival analysis of striatal injected mice shown in Fig S5F were performed on female mice. In all other experiments, groups of mixed sex were used. To reduce animal numbers, parts of the behavioral experiments in Figs S6E–G and S9E–G were done in parallel with N = 9–10 WT, EYFP, and N = 6 R6/2; EYFP mice serving as controls for both wtHDGF- and cyt-HDGF-injected mice. CAG repeat length was determined by Laragen, Inc., and amounted to 207 ± 12 (SEQ CAG No., mean ± SD) for R6/2 mice.

## Recombinant protein production

BDNF was purchased from R&D Systems (248-BD/CF). Recombinant HDGF protein was produced at the Protein Production Core Facility, Max Planck Institute of Biochemistry. In brief, pET-17b HDGF-Flag-TEV-His and cytHDGF-Flag-TEV-His were transformed into Rosetta

(DE3) cells and expressed via autoinduction (Studier, 2005) at 20°C overnight. After pelleting, the cells were resuspended in His Binding Buffer (50 mM sodium phosphate pH 8, 500 mM NaCl, 10 mM imidazole, 1 mM TCEP, and 10% glycerol) supplemented with protease inhibitor mix (prepared in-house; 1 mM AEBSF HCl, 2 $\mu$g/ml aprotinin, 1 $\mu$g/ml leupeptin, 1 $\mu$g/ml pepstatin), 2.4 U/ml benzonase (produced in-house), and 2 mM $MgCl_2$. Cell disruption was conducted using an Emulsiflex C5 homogenizer (Avestin). Lysates were centrifuged for 30 min at 50,833$g$, 4°C. Per 50 ml supernatant, 2.5 ml of 5% polyethyleneimine solution was added for nucleic acid removal, followed by stirring for 10 min, and centrifugation for 30 min at 50,833$g$, 4°C. Proteins were precipitated by stirring for 1 h with 21.8$g$ ammonium sulfate per 50 ml supernatant. After centrifugation for 30 min at 50,833$g$, 4°C, the pellet was resuspended in 50 ml His Binding Buffer and applied to a 1-ml BabyBio Ni-NTA column (45 655 103; Bio-Works) equilibrated in His Binding Buffer. The column was washed with 4% His Elution Buffer (50 mM sodium phosphate pH 8, 500 mM NaCl, 250 mM imidazole, 1 mM TCEP, and 10% glycerol), followed by protein elution in a straight gradient from 4% to 100% His Elution Buffer. The purest fractions were identified via SDS–PAGE and pooled accordingly. His-tag cleavage was performed by adding His-tagged TEV protease (produced in-house) and dialysis in PBS (137 mM NaCl, 2.7 mM KCl, 8.1 mM $Na_2HPO_4$, 1.5 mM $KH_2PO_4$, pH 7.4) with 1 mM TCEP overnight. The remaining tag and protease were removed by reverse Ni-NTA purification in batch mode (Ni-Sepharose High Performance; 17-5268-02; GE Healthcare). Final polishing of the target proteins was done through size exclusion on a HiLoad 26/600 Superdex 75 prep grade column (GE Healthcare) eluting in PBS with 1 mM DTT. Proteins were concentrated to 1 mg/ml using an Amicon Ultra 15 column (Amicon). The quality of recombinant proteins was checked via SDS–PAGE and DLS analysis. Identity was confirmed through LC-MS (micro-ToF). Protein aliquots were snap frozen in liquid nitrogen and stored at –80°C.

## PC12 cell culture and LDH assay

Stable PC12 cell lines with inducible expression of EGFP-fused HTT-exon1-Q23 or HTT-exon1-Q74 were a gift from David Rubinsztein and cultured as described (Wyttenbach et al, 2001). For viability studies, cells were seeded on coverslips in 24-well plates and transfected 12 h later using Lipofectamine LTX with Plus Reagent (15338100; Thermo Fisher Scientific) following the manufacturer's instructions. Induction of HTTQ23-EGFP or HTTQ74-EGFP was carried out 5 h after transfection by adding doxycycline at 1 $\mu$g/ml. After induction with doxycycline, cells were kept at 1% horse serum to maintain them in a quiescent-like state. LDH assay was performed with 50 $\mu$l of the medium taken at 60 h post-transfection according to the manufacturer's instructions (Pierce LDH Cytotoxicity Assay Kit; Thermo Fisher Scientific).

## Primary neuronal cultures

Cultureware was coated with 1 mg/ml poly-D-lysine (P7886; Sigma-Aldrich) in borate buffer (50 mM boric acid and 12.5 mM sodium tetraborate, pH 8.5) for 4 h to overnight at 37°C, 5% $CO_2$. After washing three times with Dulbecco's Phosphate Buffered Saline with calcium and magnesium (DPBS$_{Ca2+Mg2+}$; D8662; Sigma-Aldrich),

5 $\mu$g/ml laminin (23017-015; Gibco) in DPBS$_{Ca2+Mg2+}$ was applied for 2–4 h at 37°C, 5% $CO_2$. Meanwhile, a pregnant CD-1 female was euthanized by cervical dislocation at embryonic day 15.5. The uterus was dissected and washed in ice-cold DPBS$_{Ca2+Mg2+}$. Embryos were harvested and decapitated in ice-cold dissection medium containing 1x penicillin/streptomycin, 10 mM HEPES pH 7.4, and 10 mM $MgSO_2$ in HBSS (24020-091; Gibco). Brains were extracted and meninges were removed before dissection of neocortices. Digestion of collected neocortices was conducted at 37°C for 15 min in pre-warmed trypsin-EDTA solution (T4299; Sigma-Aldrich) with 7.5 $\mu$g/ml DNase I (10104159001; Roche). Trypsin activity was blocked by washing with pre-warmed neurobasal medium (21103-049; Gibco) supplemented with 5% heat-inactivated FBS (S0115; Biochrom). After washing with pre-warmed culture medium containing 1x B-27 (17504-044; Gibco), 1x penicillin/streptomycin and 2 mM L-Glutamine (25030-024; Gibco) in neurobasal medium, the cells were dissociated in pre-warmed culture medium by triturating and pelleted by centrifugation at 130$g$ for 5 min. Meanwhile, coated plates were washed twice with DPBS$_{Ca2+Mg2+}$. Cells were resuspended and plated in 100-mm culture dishes (Western blot), 24-well plates with cover glasses (immunocytochemistry) or 96-well plates (MTT assay) in pre-warmed culture medium at a density of 60,000/cm$^2$. Neuronal cultures were maintained at 37°C, 5% $CO_2$.

## Transfection of primary neurons

Neurons were transfected at day in vitro (DIV) 7 using CalPhos Mammalian Transfection Kit (631312; Takara Bio). Transfection solution was prepared by adding 1.5 $\mu$g DNA per construct (3 $\mu$g in total) in 200 mM $CaCl_2$ dropwise to 2x HEPES-Buffered Saline at a ratio of 1:1, followed by incubation for 30 min. Cells on cover glasses were equilibrated in a fresh pre-warmed culture medium and incubated with 30 $\mu$l transfection solution for 3 h at 37°C, 5% $CO_2$. Fresh culture medium was acidified for at least 30 min at 37°C, 10% $CO_2$ before transfer of transfected cells and incubation for 30 min at 37°C, 5% $CO_2$. Cells on cover glasses were transferred back to the original medium and incubated at 37°C, 5% $CO_2$ for protein expression.

## Lentivirus preparation and transduction of primary neurons

Lenti-X 293T cells (632180; Takara Bio) were expanded until 80% confluency in a three-layer 525 cm$^2$ culture flask (353143; Corning) at 37°C, 5% $CO_2$ in DMEM (41965-039; Gibco) supplemented with 10% heat-inactivated FBS, 1% geneticin (10131-019; Gibco), 1% nonessential amino acids solution (11140-050; Gibco), and 10 mM HEPES pH 7.4. For virus production, cells were seeded in growth medium, containing 10% FBS, 1% nonessential amino acids, and 10 mM HEPES pH 7.4 in DMEM, at a split ratio of 1:2 in a new three-layer 525 cm$^2$ culture flask. The next day, transfection mix was prepared by combining 59.5 $\mu$g of the respective pFhSynW2 expression plasmid, 35.2 $\mu$g psPAX2, and 20.5 $\mu$g pcDNA3.1 VSV-G in 4.9 ml DMEM with 345 $\mu$l TransIT-Lenti Transfection Reagent (MIR 6600; Mirus Bio) added to 4.8 ml DMEM, followed by incubation for 20 min. Growth medium was exchanged and transfection mix was added to cells. After overnight incubation, the medium was replaced by fresh pre-warmed growth medium, collected 48–52 h later, and centrifuged at

1,200$g$ for 10 min. Lentiviral supernatant was filtered through a 0.45-$\mu$m filter and concentrated using Lenti X Concentrator (631232; Takara Bio) according to the manufacturer's instructions. Virus was suspended in 600 $\mu$l buffer containing 50 mM Tris-HCl pH 7.8, 130 mM NaCl, 10 mM KCl, and 5 mM MgCl$_2$. Single-use aliquots were stored at −80°C. Viral titers were estimated using Lenti-X GoStix Plus (631280; Takara Bio) following the manufacturer's instructions using a reference value of 6.014 (TU/ml)/GV. Calculated titers were in the range of 1.35 × 10$^6$–2.77 × 10$^6$ TU/ml.

For transduction of primary neurons, 50 $\mu$l of the culture medium was removed and lentiviral vectors were added with 100 $\mu$l fresh pre-warmed culture medium at DIV 7. The amount of virus was adjusted according to the titer ranging from 0.25 to 0.50 $\mu$l.

## Immunocytochemistry on primary neurons

Cells were fixed with 4% PFA in PBS (sc-281692; ChemCruz) for 20 min. After washing twice with PBS, 50 mM NH$_4$Cl in PBS was applied for 10 min. Cells were rinsed with PBS and permeabilized with 0.3% Triton X-100 in PBS for 5 min, followed by washing 5 min with PBS. To prevent unspecific antibody binding, cells were incubated in a blocking solution containing 0.2% BSA (8076; Carl Roth), 5% normal donkey serum (017-000-121; Jackson ImmunoResearch Laboratories), 0.2% L-lysine hydrochloride, 0.2% glycine, and 0.02% NaN$_3$ in PBS for 30 min. Primary antibodies were applied for 1 h in primary antibody solution (0.3% Triton X-100, 2% BSA, and 0.02% NaN$_3$ in PBS). The following primary antibodies were used: goat anti-mCherry (AB0040-200, 1:500; Origene), mouse anti-His (DIA-900-100, 1:1,000; Dianova), chicken anti-EGFP (A10262, 1:1,000; Invitrogen), mouse anti-Flag (TA50011, 1:1,000; Origene), goat anti-Flag (ab1257, 1:2,000; Abcam), and rabbit anti-cleaved Caspase-3 (9661, 1:500; Cell Signaling Technology). Cells were washed for 5 min in PBS and incubated for 30 min in the dark with Alexa Fluor 488, Cyanine Cy3, and/or Alexa Fluor 647-conjugated secondary antibodies derived from donkey (Jackson ImmunoResearch Laboratories) at 1:250 dilution and 0.5 $\mu$g/ml DAPI in a secondary antibody solution containing 0.3% Triton X-100, 3% normal donkey serum, and 0.02% NaN$_3$ in PBS. After three 5-min washes with PBS, coverslips were dipped in Milli-Q water and mounted with ProLong Glass Antifade Mountant (P36984; Invitrogen). Images were acquired with a Leica TCS SP8 confocal microscope. For viability analysis, cells negative for active caspase-3 with intact nuclear morphology were categorized as viable.

## MTT assay

Medium was exchanged with a 5:1 mixture of pre-warmed culture medium and 5 mg/ml MTT (M5655; Sigma-Aldrich) in PBS for a total volume of 120 $\mu$l. After incubation for 3–4 h at 37°C and 5% CO$_2$, 100 $\mu$l solubilizer containing 10% SDS and 45% dimethylformamide, adjusted to pH 4.5 with acetic acid, was added. Crystals were dissolved at 37°C, 5% CO$_2$ for 4 h to overnight, and absorbance was measured at 570 nm.

## iPSC culture and differentiation to NPCs

Control iPSCs (Q33, ND36997) were obtained from NINDS Human Cell and Data Repository through the Coriell Institute. The HD Q71-iPSC line was a gift from George Q. Daley and is established and fully characterized for pluripotency (Park et al, 2008). iPSCs were cultured on Geltrex (Thermo Fisher Scientific) using mTeSR1 media (Stem Cell Technologies). Undifferentiated iPSCs were passaged using Accutase (1 U/ml; Invitrogen). iPSC lines were tested for mycoplasma contamination at least once every 2 wk confirming the absence of mycoplasma. NPCs were generated by inducing neural differentiation of iPSCs with STEMdiff Neural Induction Medium (Stem Cell Technologies) after the monolayer culture method (Chambers et al, 2009). Briefly, iPSCs were rinsed once with PBS, followed by the addition of 1 ml Gentle Dissociation Reagent (Stem Cell Technologies). After incubation for 10 min, the cells were gently collected, 2 ml of DMEM/F12 (Thermo Fisher Scientific) with 10 $\mu$M ROCK inhibitor (Abcam) was added, and cells were centrifuged at 300$g$ for 10 min. After centrifugation, the cells were resuspended on STEMdiff Neural Induction Medium with 10 $\mu$M ROCK inhibitor and plated on poly-ornithine (15 $\mu$g/ml)/laminin (10 $\mu$g/ml)-coated plates at a density of 200,000 cells/cm$^2$ for neural differentiation.

## Exogenous HDGF treatment and proteasome inhibition in NPCs

60% confluent NPCs were treated with 250 ng/ml recombinant HDGF protein or PBS as control for 24 h. The next day, cells were treated with 250 ng/ml fresh recombinant HDGF protein or PBS for 8 h in the presence of 5 $\mu$M MG-132 for proteasome inhibition or DMSO as control treatment.

## Immunocytochemistry on NPCs

Cells were fixed with 4% PFA in PBS for 20 min and permeabilized with 0.2% Triton X-100 in PBS for 10 min. After permeabilization, the cells were blocked with 3% BSA in 0.2% Triton X-100 in PBS for 10 min. The cells were incubated with mouse anti-polyQ (MAB1574, 1:50; Millipore) primary antibody for 2 h. After washing with 0.2% Triton-X in PBS, the cells were incubated with goat anti-mouse Alexa Fluor 488 (A-11029, 1:500; Thermo Fisher Scientific) secondary antibody and 2 $\mu$g/ml Hoechst 33342 (1656104; Life Technologies) for 1 h. The cells were washed with 0.2% Triton-X in PBS and with distilled water. The coverslips were mounted with FluorSave reagent (Merck Millipore).

## Filter trap assay

Cells were lysed in a non-denaturing lysis buffer supplemented with EDTA-free protease inhibitor cocktail (Roche) on ice. Cell lysates were homogenized through a 27G syringe needle. The protein concentration was determined from the whole protein lysate with a standard BCA protein assay. The equilibrated whole lysates were centrifuged at 8,000$g$ for 5 min at 4°C. The pellets were resuspended with 2% SDS and loaded onto a cellulose acetate membrane assembled in a slot blot apparatus (Bio-Rad). The membrane was washed with 0.2% SDS and retained SDS-insoluble mHTT aggregates were detected with mouse anti-polyQ antibody (MAB1574, 1:5,000; Millipore).

## Stereotactic viral injections

Viral AAV8 vectors were produced by the Viral Vector Production Unit, Universitat Autònoma de Barcelona. For striatal injections, R6/2 mice and their littermates underwent stereotactic surgery at 4 wk of age. Mice were injected intraperitoneally with 15 ml/kg body

weight (BW) 20% mannitol in 0.9% NaCl. Metamizol (200 mg/kg BW) was orally administered. Anesthesia was induced with 4% isoflurane and maintained at 1.5–2% isoflurane using an $O_2$ flow rate of 1 liter/min. Body temperature was maintained with a heating pad. Carprofen (5 mg/kg BW) was given subcutaneously. Per injection site, mice were injected with $10^9$ gc AAV8 in 0.2 $\mu$l with a final concentration of 6% mannitol in 0.9% NaCl at (X, Y, Z) = (±1.7, 1.0, −3.0) and (X, Y, Z) = (±2.1, 0.3, −3.0) mm from bregma. For each of the four injections, the glass capillary was left in position for 3 min. Skull holes were covered with bone wax and the incision was closed with sutures. Intracerebroventricular viral injections at postnatal day 0 (P0) were performed similarly as previously described (Kim et al, 2014). In brief, pregnant females were monitored for birth at least every 12 h starting 17 d after the plug date to ensure surgery of newborn pups within 24 h after birth. Anesthesia was induced with 5% isoflurane and maintained at 2% isoflurane using an $O_2$ flow rate of 1 liter/min. Xylocaine 2% jelly (6077215.00.00; Aspen) was applied on the prospective injection sites for local anesthetic blockade. Per hemisphere, $10^{10}$ gc AAV8 in 1–2 $\mu$l was stereotactically injected with 15 nl/sec using a microinjection system (Nanoliter 2010, WPI) at (X, Y, Z) = (±0.8, 1.5, −1.5) mm from lambda. Xylocaine 2% jelly was again applied on injection sites to seal wounds. During surgery and for recovery, neonates were kept on a warming pad. To ensure success of fostering with CD-1 females, fecal pellets from the foster mother were solved in water and rubbed on the back of the injected pups. Neonates were placed in the foster mother's cage as soon as they recovered and were moving normally. Mice were transferred to an inverted light cycle after surgery.

## Behavior and life span analysis

All behavioral assessments were conducted during the dark phase of the diurnal cycle. For the open field test, mice were video recorded while exploring a custom-made squared box (40 × 40 × 40 cm) with black walls and white floor for 10 min with lights on. EthoVision XT 14 software (Noldus Information Technology) was used for automated tracking to quantify the distance traveled. Rotarod analysis was conducted on a RotaRod NG (Ugo Basile). Mice were trained twice on two consecutive days at 5 rpm for 5 min. On the third day, the latency to fall was measured with accelerating speed from 5 to 40 rpm over a 5 min period, and averaged over three trials. Forepaw grip strength was determined using the BIO-GS3 Grip Strength Test (Bioseb) with the BIO-GRIPBS bar for mice (Bioseb) as grasping tool. Measurements were averaged from three consecutive trials. For survival analysis, endpoint measures were severe burden according to behavior, appearance, and body weight or loss of righting reflex.

## Immunostaining and fluorescence in situ hybridization on mouse tissue

Mice were transcardially perfused with PBS for 4 min, followed by 4% PFA in PBS for 6 min at 3–3.5 ml/min under ketamine/xylazine anesthesia. Brains and, if indicated, spinal cords were extracted and postfixed overnight in 4% PFA in PBS at 4°C.

For immunostaining of free-floating sections, fixed tissue was embedded in 4% agarose in PBS and serial 70-$\mu$m-thick sections were cut in PBS with a vibratome. Sections were permeabilized with 0.5% Triton X-100 in PBS for 15 min. If indicated, antigen retrieval was performed in 10 mM trisodium citrate pH 6 with 0.05% Tween 20 at 80°C for 15 min at 300 rpm in an Eppendorf ThermoMixer. To prevent unspecific antibody binding, sections were incubated in the blocking solution for 1 h on a shaker. Primary antibodies were applied in the primary antibody solution overnight at 4°C on a shaker. The following primary antibodies were used: rabbit anti-HDGF (ab128921, 1:500; Abcam), mouse anti-HTT (EM48, MAB5374, 1:500; Chemicon), mouse anti-Flag (TA50011, 1:1,000; Origene), goat anti-ChAT (AB144, 1:500; Chemicon), goat anti-DARPP32 (LS-C150127, 1:300; LifeSpan Biosciences), mouse anti-NGRN (MAB7947, 1:60; R&D Systems), chicken anti-GFAP (AP31806PU-N, 1:2,000, with antigen retrieval; Origene), mouse anti-APC (OP80, 1:20, with antigen retrieval; Calbiochem), and goat anti-IBA1 (ab107159, 1:1,000, with antigen retrieval; Abcam). After three 10-min washes in PBS, sections were incubated for 1 h in the dark with Alexa Fluor 488, Cyanine Cy3, and/or Alexa Fluor 647-conjugated secondary antibodies derived from donkey at 1:250 dilution in the secondary antibody solution with gentle shaking. NeuroTrace 640/660 (N21483; Invitrogen) was added at 1:500 dilution to the secondary antibody solution if indicated. Sections were washed three times for 10 min in PBS with DAPI added in the middle washing step at a concentration of 0.5 $\mu$g/ml. ProLong Glass Antifade Mountant was used for mounting.

For fluorescence in situ hybridization, fixed brains were immersed in 15% sucrose in Dulbecco's Phosphate Buffered Saline (DPBS; D8537; Sigma-Aldrich), followed by 30% sucrose in DPBS, until the tissue sunk. Cryopreserved brains were frozen in Tissue-Tek O.C.T. Compound (4583; Sakura Finetek) and coronally sectioned at 10 $\mu$m thickness on a cryostat. RNAscope Fluorescent Multiplex Assay (320850, 322000, and 322340; ACD) was conducted according to the manufacturer's instructions (320293-USM and 320535-TN; ACD) with RNAscope Probe against *Hdgf* (524601; ACD). Immunostaining against HDGF protein was performed before counterstaining with DAPI similar as immunostaining of free-floating sections. Briefly, brain sections were incubated in the blocking solution for 2 h, followed by overnight incubation with rabbit anti-HDGF antibody (ab128921, 1:500; Abcam) in the primary antibody solution at 4°C. After washing four times for 5 min with RNAscope 1x Wash Buffer, Alexa Fluor 488-conjugated secondary antibody raised in donkey was applied at 1:250 dilution in the secondary antibody solution for 2 h. Sections were washed four times for 5 min with RNAscope 1x Wash Buffer, counterstained with RNAscope DAPI, and mounted with ProLong Glass Antifade Mountant. Images were acquired with a Leica TCS SP8 confocal microscope.

## Western blotting

Primary neurons were lysed in 100 $\mu$l ice-cold lysis buffer containing 50 mM Tris–HCl pH 7.5, 150 mM NaCl, 1% Triton-X 100, 2 mM EDTA, protease inhibitor cocktail (04693159001; Roche), and phosphatase inhibitor (04906837001; Roche). Lysates were centrifuged for 10 min at 4,000$g$, 4°C, and supernatants were collected for Western blotting.

To obtain mouse brain tissue lysates, mice were euthanized by cervical dislocation. Brains were extracted and cerebellum,

hippocampus, cortex, and striatum (per hemisphere) were dissected on ice. Dissected brain regions were homogenized in ice-cold lysis buffer, incubated on ice for at least 30 min, and centrifuged at 15,000$g$ for 15 min at 4°C. Supernatants were collected and protein concentrations were determined with DC Protein Assay Kit (5000112; Bio-Rad) following the manufacturer's instructions.

Samples were boiled at 95°C for 5 min with Laemmli sample buffer (1610747; Bio-Rad) containing 2-mercaptoethanol. Per lane, 25 $\mu$l cell lysate and 50 $\mu$g protein for tissue lysates was loaded on 4–15% (5678084; Bio-Rad) and 10% TGX Stain-Free Protein Gels (4568034 and 5678034; Bio-Rad), respectively. Proteins were separated along with 5 $\mu$l Precision Plus Protein All Blue Prestained Protein Standards (1610373; Bio-Rad) at 80–120 V in SDS–PAGE-running buffer containing 25 mM Tris, 192 mM glycine, and 0.1% SDS. After electrophoretic separation, stain-free gels were activated using the ChemiDoc MP Imaging System (17001402; Bio-Rad). Transfer onto low-fluorescence polyvinylidene fluoride membranes (1620264; Bio-Rad) was conducted at 2.5 A, up to 25 V for 10 min with the Trans-Blot Turbo Transfer System (1704150; Bio-Rad) according to the manufacturer's instructions for using traditional semi-dry consumables. Once the protein transfer was complete, the stain-free blot image was acquired using the ChemiDoc MP imager. Membranes were then blocked for 1 h with 3% BSA and 5% powdered milk (T145; Carl Roth) in TBS (20 mM Tris-HCl pH 8, 150 mM NaCl) with 0.1% Tween 20 (TBS-T). After rinsing twice and washing for 5 min in TBS-T, primary antibodies were applied overnight at 1:1,000 dilution in 3% BSA and 0.01% NaN$_3$ in TBS-T at 4°C. The following primary antibodies were used: rabbit anti-HDGF (244498; Abcam), mouse anti-AKT (2920; Cell Signaling Technology), rabbit anti-p-AKT (4060; Cell Signaling Technology), rabbit anti-ERK (9102; Cell Signaling Technology), and mouse anti-p-ERK (9106; Cell Signaling Technology). The next day, membranes were rinsed twice and washed three times for 10 min in TBS-T. Rhodamine-conjugated anti-tubulin antibody (12004166; Bio-Rad) and Star-Bright Blue 520/700 secondary antibodies (Bio-Rad) were applied at 1:2,500 dilution in TBS for 1 h, followed by rinsing twice and washing three times for 10 min in TBS-T. Blot image was acquired using the ChemiDoc MP imager and quantified with Image Lab version 6.0.1, build 34 software (Bio-Rad). HDGF protein quantity was normalized to total protein per sample. For quantification of exogenous HDGF, the area under the curve was quantified at the band heights of EYFP-P2A-Flag-HDGF/cyt-HDGF and Flag-HDGF/cyt-HDGF irrespective of the presence or absence of a protein band. The sum was calculated and normalized to total protein per sample.

### Immunohistochemistry on human brain sections

Formalin-fixed paraffin-embedded tissue sections of 5 $\mu$m thickness from the primary motor cortex of three HD autopsy cases and three age-matched controls were provided by the Neurobiobank Munich, Ludwig-Maximilians-Universität München. Informed consent was available for all cases. The experiments were approved by the ethics committee of the Max Planck Society and performed in accordance with the relevant guidelines and regulations. All HD cases were symptomatic; demographic information is described in Burgold et al (2019).

Immunohistochemistry was performed on a VENTATA BenchMark ULTRA (Roche). After standard pretreatment in CC1 buffer (Roche), sections were incubated with rabbit anti-HDGF antibody (ab244498; Abcam) at a dilution of 1:100 for 32 min. The UltraView Universal DAB Detection Kit (Roche) was used for detection and counterstaining was performed with hematoxylin for 4 min. Images were acquired with a Leica THUNDER imager. Control stainings without the primary antibody were imaged using an Olympus BX45 microscope with Olympus DP32 camera.

### Data analysis and statistics

Images were analyzed and/or processed with the open-source image analysis software Fiji (Schindelin et al, 2012). GraphPad Prism 9.2.0 (GraphPad Software) was used for graphical representation and statistical analysis. Statistics are detailed in the figure legends. Differences were considered statistically significant with $P < 0.05$. In bar plots, columns with error bars represent mean ± SD. In violin plots, the thick black line and colored area indicate the median and interquartile range, respectively.

# Supplementary Information

# Acknowledgements

We thank Nicole Martin for help with biochemistry and histology; Maximilian Gantz and Florian Leiß-Maier for help with histology and recombinant HDGF protein production; Magdalena Böhm for technical assistance and mouse genotyping; Andrea Wevers for helpful discussions; Henry Klein for help with image analysis; Elke Franke for assistance with cell culture; Azra Fatima for initial experiments with NPCs; Leopold Urich for recombinant protein production; and Michael Schmidt for assistance with immunohistochemistry. This work was funded by the European Research Council (ERC) Synergy Grant under FP7 GA number ERC-2012-*SyG_318987*-Toxic Protein Aggregation in Neurodegeneration (ToPAG) (to R Klein); the German Research Foundation (Deutsche Forschungsgemeinschaft, DFG) (grant number VI742/4-1 to D Vilchez) and by the Max Planck Society for the Advancement of Science. D Vilchez and I Dudanova acknowledge funding from the DFG through Germany's Excellence Strategy—CECAD, EXC 2030–390661388. We acknowledge support for the Article Processing Charge from the DFG (491454339).

### Author Contributions

K Voelkl: conceptualization, formal analysis, investigation, visualization, methodology, and writing—original draft, review, and editing.
S Gutiérrez-Ángel: conceptualization, formal analysis, investigation, and methodology.
S Keeling: formal analysis and investigation.
S Koyuncu: formal analysis and investigation.
M da Silva Padilha: formal analysis and investigation.
D Feigenbutz: investigation.
T Arzberger: resources.
D Vilchez: supervision and writing—review and editing.

R Klein: conceptualization, supervision, funding acquisition, and writing—review and editing.

I Dudanova: conceptualization, supervision, project administration, and writing—original draft, review, and editing.

## Conflict of Interest Statement

The authors declare that they have no conflict of interest.

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
