## [Reviewer comments · Life Science Alliance]

Life Science Alliance

Neuroprotective effects of hepatoma-derived growth factor in models of Huntington's disease

Kerstin Voelkl, Sara Gutiérrez-Ángel, Sophie Keeling, Seda Koyuncu, Miguel da Silva Padilha, Dennis Feigenbutz, Thomas Arzberger, David Vilchez, Rüdiger Klein, and Irina Dudanova

DOI: <https://doi.org/10.26508/lsa.202302018>

Corresponding author(s): Irina Dudanova, University Hospital Cologne and Rüdiger Klein, Max Planck Institute for Biological Intelligence

Review Timeline:

Submission Date:	2023-02-28
Editorial Decision:	2023-03-27
Revision Received:	2023-07-06
Editorial Decision:	2023-07-24
Revision Received:	2023-07-29
Accepted:	2023-07-31

Transaction Report:

March 27, 2023

Re: Life Science Alliance manuscript #LSA-2023-02018

Prof. Irina Dudanova
University Hospital Cologne
Center for Anatomy
Kerpener Str. 62
Cologne 50937
Germany

Dear Dr. Dudanova,

Thank you for submitting your manuscript entitled "Neuroprotective effects of hepatoma-derived growth factor in models of Huntington's disease" to Life Science Alliance. The manuscript was assessed by expert reviewers, whose comments are appended to this letter. We invite you to submit a revised manuscript addressing the Reviewer comments.

Thank you for this interesting contribution to Life Science Alliance. We are looking forward to receiving your revised manuscript.

Sincerely,

B. MANUSCRIPT ORGANIZATION AND FORMATTING:

Reviewer #1 (Comments to the Authors (Required)):

Voelkl et al, investigate the neuroprotective potential of HDGF in Huntington's disease (HD). They have employed various model systems of HD ranging from immortalized cell lines to rodent models and human patient iPSC-derived NPCs to examine the role of HDGF in modulating disease kinetics. This is highly applaudable. They show with dual immunofluorescence and fluorescent in situ hybridization that neurons endogenously express HDGF and that the expression levels of HDGF inversely correlates with cell type-specific vulnerability in HD. Crossing R6/2 mice with HDGF^{-/-} mice led to a significant reduction in their life span, suggesting a probable role for HDGF in pathology modulation. Nevertheless, AAV-mediated delivery of HDGF into the brain despite reducing in mutant mice, the Huntingtin aggregate load, had no significant ameliorative effect on motor symptoms or lifespan. Both forms of HDGF (nuclear or cytoplasmic) were shown to be equally efficient in providing neuroprotection in cell cultures assays. To further assess this cytoprotective potential, importantly they applied recombinant HDGF protein to neural progenitor cells differentiated from human patient-derived induced pluripotent stem cells (iPSCs), revealing normalized nuclear size and reduced mutant Huntingtin aggregation, indicative of a role of HDGF in reducing/resolving HD inclusion load. Overall, the study is performed to a very high standard and is of definite interest to the field. I enthusiastically recommend the publication of this study after suggested revisions.

Major comments:

- 1) The ability of HDGF to reduce aggregation load is of high interest to the neurodegeneration community. Authors should investigate whether this ability is linked to improved cell clearance of the inclusions (proteasome, autophagy). Autophagic flux as well as proteasome function should be assessed in cell culture assays.
- 2) The ablation of HDGF accelerates the disease phenotype is an interesting observation, however, no further analyses was done at a cellular level. Is there an alteration in the temporal kinetics of aggregate formation.
- 3) It remains unclear, whether anticipated loss of affected neurons within distinct brain regions is the main cause for reduced lifespan in HD/HDGF^{-/-} mice or is it due to some other unrelated cause.
- 4) The human patient immunohistochemistry data seems a bit meaningless with no quantification to back it up. HDGF intensity as well as neuron numbers would help to put this data into context of the study. Is it possible that those neurons which present normal unchanged levels of HDGF expression are likely surviving neurons.
- 5) A dual labelling for HDGF and HD inclusions in human patient tissue would be informative, as to understanding whether those neurons with normal HDGF levels lack/reduced HD inclusions.
- 6) The experiments performed on patient iPSC-derived NPCs are of high value. Besides, showing a reduction in HD inclusion, survival assays, either by challenging the NPCs with a stressor or blocking proteasome or autophagy in the presence of HDGF would be important additions.

Reviewer #2 (Comments to the Authors (Required)):

Voelkl et al. have investigated the effect of HDGF on the cellular and behavioral phenotype of Huntington's disease mouse models. They find that HDGF rescues the phenotype of HTT model cell lines and that deletion of HDGF in HTT mice mildly worsens the phenotype and shortens the lifespan. However, overexpression of HDGF in HTT model mice, neither after juvenile nor after newborn transduction, has any effects on the HTT phenotype. They further find that the cellular effects were equally effectively produced by both the nuclear as well as the cytoplasmic HDGF. Enormous amount of work is invested into these studies, experiments appear well designed and carefully conducted and the results are clearly reported and interesting and certainly worth publishing. Unfortunately, the fact that nearly nothing is known about the mechanism of HDGF action essentially precludes any mechanistic studies that could have increased the impact further, but this is not the fault of the current authors. There are a number of things that the authors still could do, and I am sure they are aware of that, but I think that the manuscript

in the present form nicely wraps up what needs to be known about this topic and I therefore do not feel that there is any substantial additional work that needs to be done. Below, I have listed some minor issues that the authors may want to include in a potential revised version:

1. I think that the issue of reverse correlation between the expression pattern of endogenous HDGF and the vulnerability to HD is overstated. The authors suggest that the greater resilience of cerebellum to HD correlates with higher HD expression. However, cerebellum is so tightly packed with cerebellar granule cells that one certainly expects that a nuclear factor expressed in CGNs would appear as a stronger band in a Western from cerebellum than from much less neuron-dense cortex and striatum, even though the expression levels per cell were comparable. Although Fig 2a does not zoom into cerebellum (plus the irritating habit of the authors to always co-stain with HDGF and DAPI that overlap and more or less preclude any useful examination of HDGF levels), one gets the impression that the major source of HDGF in the cerebellum are indeed CGNs. In this context it is interesting to note that figures S5a-b and S8a suggest that the externally expressed HDGF is largely confined to Purkinje neurons in the cerebellum. In addition, the rest of localization is so cursory that I do not think that it is a strong basis for correlating expression levels with cellular vulnerability. I therefore suggest that the authors would tone down their conclusion of an inverse correlation between the expression levels and vulnerability, as now stated in the abstract and elsewhere.
2. Figure 8a and 8b both have PBS as one treatment, but the outcomes are different. I have understood that these two PBSs represent different treatments; if so, please change the designation on either one so that there is no confusion.
3. Further on Figure 8, c shows three images that apparently are photos of a slot-blot filter. Are the 3 "lanes" duplicates of the same experiment or somehow different? If they are duplicates, it is remarkable that the blot on the right-hand side fail to show any effect of the MG treatment and, consequently also of the different forms of HDGF. Are these the only 3 replicates and were they quantitated in some way? I understood that the quantitation shown in S10b represents the immunostaining data. The figure 8c could be better labeled so that the reader immediately grasps what is being shown.
4. Discussion, p. 18, 2nd paragraph: The authors suggest that the failure of HDGF overexpression in the brain to rescue the HTT phenotype would be caused by the effects of HDGF in peripheral organs. Although this, indeed, is consistent with the data, the assays employed, rotarod and open field, hardly reflect HDGF expression in heart or muscle. Another explanation, related to the first speculation by the authors, is that the rather strong neuronal HDGF expression already saturates the putative receptors/signaling pathways, therefore, one would not expect any additional benefit. As for the first speculation, this one remains speculation until the signal transduction of HDGF has been clarified.

In conclusion, I would like to congratulate the authors for nice work.

Point-by-point response to the reviewers' comments

The changes made in the manuscript text in response to the reviewers' comments are highlighted in yellow.

Reviewer 1

Voelkl et al., investigate the neuroprotective potential of HDGF in Huntington's disease (HD). They have employed various model systems of HD ranging from immortalized cell lines to rodent models and human patient iPSC-derived NPCs to examine the role of HDGF in modulating disease kinetics. This is highly applaudable. They show with dual immunofluorescence and fluorescent in situ hybridization that neurons endogenously express HDGF and that the expression levels of HDGF inversely correlates with cell type-specific vulnerability in HD. Crossing R6/2 mice with HDGF^{-/-} mice led to a significant reduction in their life span, suggesting a probable role for HDGF in pathology modulation. Nevertheless, AAV-mediated delivery of HDGF into the brain despite reducing in mutant mice, the Huntingtin aggregate load, had no significant ameliorative effect on motor symptoms or lifespan. Both forms of HDGF (nuclear or cytoplasmic) were shown to be equally efficient in providing neuroprotection in cell cultures assays. To further assess this cytoprotective potential, importantly they applied recombinant HDGF protein to neural progenitor cells differentiated from human patient-derived induced pluripotent stem cells (iPSCs), revealing normalized nuclear size and reduced mutant Huntingtin aggregation, indicative of a role of HDGF in reducing/resolving HD inclusion load. Overall, the study is performed to a very high standard and is of definite interest to the field. I enthusiastically recommend the publication of this study after suggested revisions.

We sincerely thank the reviewer for the appreciative comments on our approaches and the overall enthusiastic evaluation of our work.

Major comments:

1) The ability of HDGF to reduce aggregation load is of high interest to the neurodegeneration community. Authors should investigate whether this ability is linked to improved cell clearance of the inclusions (proteasome, autophagy). Autophagic flux as well as proteasome function should be assessed in cell culture assays.

We thank the reviewer for the interesting suggestion. To address this question, we conducted a number of experiments in HEK293 cells. First, to test whether HDGF regulates HTT degradation, we co-expressed pathological HTT-exon1-Q97-mCherry and control HTT-exon1-Q25-mCherry with wtHDGF or cytHDGF and monitored HTT protein levels upon inhibition of protein synthesis with Cycloheximide. We observed lower levels of wtHTT, but not mHTT, in the presence of HDGF (Fig. R1a). In addition, we probed the ability of HDGF to regulate proteasome activity in HEK293 cells co-transfected with wtHTT or mHTT and wtHDGF or cytHDGF. wtHDGF significantly increased proteasome activity in wtHTT cells, while cytHDGF boosted proteasome activity in the presence of mHTT (Fig. R1b). We furthermore tried to analyze the proteasomal and autophagic clearance of mHTT in the presence and absence of HDGF, but due to technical difficulties unfortunately did not manage to obtain conclusive results within the timeframe of the revisions. Taken together, our new data is compatible with the scenario that HDGF increases proteasome activity and promotes proteasomal degradation of certain substrates (Fig. R1b), probably including wtHTT (Fig. R1a), however this effect does not seem to extend to mHTT. As the results were overall not conclusive enough, we have not included them into the manuscript.

Figure R1. Effect of HDGF on mHTT degradation and proteasome activity. **a**, HEK293 cells were co-transfected with the indicated constructs, treated with 50 $\mu\text{g}/\text{mL}$ Cycloheximide for the indicated duration of time, and protein quantity of HTT was measured by Western blot. HTT protein quantity was first normalized to Tubulin and then to the average value of the respective DMSO condition. N=3 independent experiments; Two-way ANOVA with Bonferroni's multiple comparison test per HTT construct. HTTQ25-mCherry, ANOVA: Cycloheximide, * $p=0.0306$; HDGF, * $p=0.0303$; Cycloheximide x HDGF, $p=0.3621$. HTTQ97-mCherry, ANOVA: Cycloheximide, *** $p=0.0009$; HDGF, $p=0.8018$; Cycloheximide x HDGF, $p=0.8937$. **b**, Quantification of chymotrypsin-like proteasome activity in lysates of HEK293 cells transfected with the indicated constructs. N=3 independent experiments; Two-way ANOVA with Bonferroni's multiple comparison test per HTT construct. ANOVA: PolyQ, **** $p<0.0001$; HDGF, *** $p=0.0002$; PolyQ x HDGF, **** $p<0.0001$. Significant pairwise comparisons are indicated on the graphs. * $p<0.05$; ** $p<0.01$; *** $p<0.001$; **** $p<0.0001$.

2) The ablation of HDGF accelerates the disease phenotype is an interesting observation, however, no further analyses was done at a cellular level. Is there an alteration in the temporal kinetics of aggregate formation.

Thank you for the helpful comment. We have quantified the fraction of neurons with inclusions as well as inclusion size in the striatum and cortex of 8-week-old R6/2 mice with and without genetic HDGF ablation. We did not detect significant changes in the percentage of neurons bearing mHTT inclusions, nor in the size of the inclusions, possibly because both parameters were already saturated in R6/2 mice at this age. The new data is described in the Results on p. 6-7, lines 177-179, and presented in Supplementary Fig. S4a-c.

Unfortunately, we were not able to address the temporal dynamics of aggregate formation at earlier ages that might have been more informative, because it was not possible to generate new cohorts of R6/2; HDGF^{-/-} animals for these analyses within the time frame of the revisions.

3) It remains unclear, whether anticipated loss of affected neurons within distinct brain regions is the main cause for reduced lifespan in HD/HDGF^{-/-} mice or is it due to some other unrelated cause.

To address this question, we have performed neuronal counts and measured the area of the striatum and cortex in 12-week-old R6/2 mice and control littermates with and without HDGF ablation. We did not observe differences in neuron numbers or in the degree of striatal and cortical atrophy. The data is described in the Results on p. 6-7, lines 177-179, and shown in Supplementary Fig. S4d-e. These results argue against additional neuronal loss in the absence of HDGF. The observed aggravation of behavioral phenotypes and reduction in life span of R6/2 mice is therefore likely due to other reasons, such as e.g. dysfunction and insufficient trophic support of the surviving neurons.

4) The human patient immunohistochemistry data seems a bit meaningless with no quantification to back it up. HDGF intensity as well as neuron numbers would help to put this data into context of the study. Is it possible that those neurons which present normal unchanged levels of HDGF expression are likely surviving neurons.

We agree with the reviewer that a quantification would be desirable. We were unfortunately not able to quantify HDGF intensity on the available immunohistochemistry images due to the suboptimal imaging settings and the overlap of blue and brown colors. We have attempted performing immunofluorescence to overcome these difficulties, but could not reliably detect HDGF. We have moved the human expression data to the supplements, since no quantitative assessment of the data was possible (now Supplementary Fig. S3a-f).

5) A dual labelling for HDGF and HD inclusions in human patient tissue would be informative, as to understanding whether those neurons with normal HDGF levels lack/reduced HD inclusions.

We agree that dual labeling for HDGF and mHTT inclusions would be very helpful. We have tried different antigen retrieval methods and various staining protocols, but were unfortunately not able to obtain double labeling for HDGF and mHTT in human brain sections. As the available data from human tissue is not very conclusive, we have moved these results to the Supplements (Supplementary Fig. S3a-f, see also previous comment).

6) The experiments performed on patient iPSC-derived NPCs are of high value. Besides, showing a reduction in HD inclusion, survival assays, either by challenging the NPCs with a stressor or blocking proteasome or autophagy in the presence of HDGF would be important additions.

We thank the reviewer for the interesting suggestion. However, analyses of viability in iPSC-derived cultures are complicated by the lack of mHTT effect on survival in this model system. The overall very low cell death rates of HD-iPSC lines that are similar to control lines were already demonstrated in a previous publication by Koyuncu et al., *Nat Commun* 2018 (Fig. 1e). As suggested by the reviewer, we have now also performed survival analyses upon differentiation of HD-iPSCs into NPCs and treatment with a proteasome inhibitor. As in iPSCs, we observed similar viability of NPCs with and without the HD mutation (Fig. R2). The potential positive effect of HDGF on survival therefore could not be reliably assessed in this system.

Figure R2. Quantification of the fraction of viable NPCs in the indicated conditions. Aggregation was induced 24 h after treatment with PBS or recombinant HDGF (250 ng/ml) by proteasome inhibition with MG-132 for 8 h. Viability was assessed by immunostaining against cleaved caspase-3. N=4 experiments. For HTTQ71: 2-way ANOVA with Bonferroni's multiple comparisons test; ANOVA: MG132, $p=0.4938$; HDGF, $p=0.1855$; MG132 x HDGF, $**P=0.0096$.

Reviewer #2

Voelkl et al. have investigated the effect of HDGF on the cellular and behavioral phenotype of Huntington's disease mouse models. They find that HDGF rescues the phenotype of HTT model cell lines and that deletion of HDGF in HTT mice mildly worsens the phenotype and shortens the lifespan. However, overexpression of HDGF in HTT model mice, neither after juvenile nor after newborn transduction, has any effects on the HTT phenotype. They further find that the cellular effects were equally effectively produced by both the nuclear as well as the cytoplasmic HDGF. Enormous amount of work is invested into these studies, experiments appear well designed and carefully conducted and the results are clearly reported and interesting and certainly worth publishing. Unfortunately, the fact that nearly nothing is known about the mechanism of HDGF action essentially precludes any mechanistic studies that could have increased the impact further, but this is not the fault of the current authors. There are a number of things that the authors still could do, and I am sure they are aware of that, but I think that the manuscript in the present form nicely wraps up what needs to be known about this topic and I therefore do not feel that there is any substantial additional work that needs to be done. Below, I have listed some minor issues that the authors may want to include in a potential revised version:

We are grateful to the reviewer for the positive evaluation of the manuscript, for appreciating our efforts on the project, and for the constructive suggestions.

1. I think that the issue of reverse correlation between the expression pattern of endogenous HDGF and the vulnerability to HD is overstated. The authors suggest that the greater resilience of cerebellum to HD correlates with higher HD expression. However, cerebellum is so tightly packed with cerebellar granule cells that one certainly expects that a nuclear factor expressed in CGNs would appear as a stronger band in a Western from cerebellum than from much less neuron-dense cortex and striatum, even though the expression levels per cell were comparable. Although Fig 2a does not zoom into cerebellum (plus the irritating habit of the authors to always co-stain with HDGF and DAPI that overlap and more or less preclude any useful examination of HDGF levels), one gets the impression that the major source of HDGF in the cerebellum are indeed CGNs. In this context it is interesting to note that figures S5a-b and S8a suggest that the externally expressed HDGF is largely confined to Purkinje neurons in the cerebellum.

In addition, the rest of localization is so cursory that I do not think that it is a strong basis for correlating expression levels with cellular vulnerability. I therefore suggest that the authors would tone down their conclusion of an inverse correlation between the expression levels and vulnerability, as now stated in the abstract and elsewhere.

Regarding the expression in the cerebellum, please note that for the Western blot quantifications we normalized HDGF levels to the total protein, therefore differences in cell density are already accounted for. Nevertheless, we agree with the reviewer that the expression pattern of endogenous HDGF and AAV-HDGF is different (Fig. 2a and S6a), and that it is difficult to make conclusions about correlation with vulnerability of brain regions. We have therefore removed the respective statements from the text, modified the scheme in Fig. 3a and rearranged Fig. 3: The Western blot data for HDGF levels in different brain regions that was previously shown in Fig. 3b-c is now part of the general characterization of HDGF expression in the central nervous system (Fig. 2b-c).

We have also rephrased and toned down the claims about the correlation of HDGF expression with vulnerability of cell types:

Abstract, lines 43-44: "We show that HD-vulnerable neuronal cell types in the striatum and cortex express lower levels of HDGF than the more resistant ones".

Results, p. 4, section title line 109: "HDGF expression in the mouse brain".

Results, p. 5, lines 138-139: "Taken together, these results suggest that higher HDGF expression in the brain might correlate with neuronal resistance to HD".

As suggested by the reviewer, we have separated the HDGF and DAPI channels in the figures showing HDGF expression, to allow for a better visual assessment of the HDGF staining. In Fig. 2a and S6a (previously S5a) HDGF staining is now shown on its own without DAPI. In Fig. 2d-f (previously 2b-d) and Fig. S2c we have introduced additional images for individual channels.

2. Figure 8a and 8b both have PBS as one treatment, but the outcomes are different. I have understood that these two PBSs represent different treatments; if so, please change the designation on either one so that there is no confusion.

Thank you for pointing out this unclarity. Fig. 7a (previously 8a) shows viability of untransfected wildtype neurons in starving (B27-free) medium, whereas Fig. 7b (previously 8b) shows viability of cells transfected with control (HTTQ25) or mutant Huntingtin (HTTQ97) in standard complete medium. We have indicated the conditions in B27-free medium with a shaded blue area on the graph in Fig. 7a to make this clear.

3. Further on Figure 8, c shows three images that apparently are photos of a slot-blot filter. Are the 3 "lanes" duplicates of the same experiment or somehow different? If they are duplicates, it is remarkable that the blot on the right-hand side fail to show any effect of the MG treatment and, consequently also of the different forms of HDGF. Are these the only 3 replicates and were they quantitated in some way? I understood that the quantitation shown in S10b represents the immunostaining data. The figure 8c could be better labeled so that the reader immediately grasps what is being shown.

The experiment was performed three times, as is now indicated above the images. There is an effect of the MG-132 treatment also in the 3rd experiment (compare first and fourth band), albeit a mild one. Likewise, the effect of HDGF treatment is less strong in this experiment. We have quantified the intensity of the bands as suggested by the reviewer, the data is now shown in Fig. 7d and described in the Results, p. 10, lines 285-289.

4. Discussion, p. 18, 2nd paragraph: The authors suggest that the failure of HDGF overexpression in the brain to rescue the HTT phenotype would be caused by the effects of HDGF in peripheral organs. Although this, indeed, is consistent with the data, the assays employed, rotarod and open field, hardly

reflect HDGF expression in heart or muscle. Another explanation, related to the first speculation by the authors, is that the rather strong neuronal HDGF expression already saturates the putative receptors/signaling pathways, therefore, one would not expect any additional benefit. As for the first speculation, this one remains speculation until the signal transduction of HDGF has been clarified.

We agree with the reviewer and have modified this paragraph in the discussion. It now reads (p. 11, lines 308-320):

“...One potential factor contributing to the modest beneficial effects of HDGF overexpression *in vivo* might be that its levels were only increased in the brain, but not in the peripheral tissues. As HTT is ubiquitously expressed, its mutation causes pathological changes in multiple tissues beyond the brain including the musculoskeletal and cardiovascular systems, which may play an important role in disease manifestations (van der Burg et al., 2009). Another possible explanation is that the relatively high endogenous expression of HDGF in the nervous system saturates the activation of its receptors and/or signaling pathways, precluding additional benefit from ectopic expression. Alternatively, increased HDGF concentration alone might not be sufficient to rescue HD phenotypes, if the downstream signaling components are limited or impaired in due to the HD mutation... The putative HDGF receptor and the downstream signaling pathway is currently unclear and remains to be explored.”

In conclusion, I would like to congratulate the authors for nice work.

Thank you!

July 24, 2023

RE: Life Science Alliance Manuscript #LSA-2023-02018R

Prof. Irina Dudanova
University Hospital Cologne
Center for Anatomy
Kerpener Str. 62
Cologne 50937
Germany

Dear Dr. Dudanova,

Thank you for submitting your revised manuscript entitled "Neuroprotective effects of hepatoma-derived growth factor in models of Huntington's disease". We would be happy to publish your paper in Life Science Alliance pending final revisions necessary to meet our formatting guidelines.

-please add ORCID ID for the corresponding author--you should have received instructions on how to do so

A. FINAL FILES:

B. MANUSCRIPT ORGANIZATION AND FORMATTING:

****It is Life Science Alliance policy that if requested, original data images must be made available to the editors. Failure to provide original images upon request will result in unavoidable delays in publication. Please ensure that you have access to all original**

data images prior to final submission.**

The license to publish form must be signed before your manuscript can be sent to production. A link to the electronic license to publish form will be sent to the corresponding author only. Please take a moment to check your funder requirements.

Sincerely,

Reviewer #1 (Comments to the Authors (Required)):

The authors have done an exemplary job of revising the manuscript.
The manuscript has substantially improved and will be an important contribution in the field of neurodegeneration. The manuscript should be published in it's current form.

Reviewer #2 (Comments to the Authors (Required)):

The changes that the authors have made to the manuscript are sufficient for me. Please publish.

July 31, 2023

RE: Life Science Alliance Manuscript #LSA-2023-02018RR

Prof. Irina Dudanova
University Hospital Cologne
Center for Anatomy
Kerpener Str. 62
Cologne 50937
Germany

Dear Dr. Dudanova,

Thank you for submitting your Research Article entitled "Neuroprotective effects of hepatoma-derived growth factor in models of Huntington's disease". It is a pleasure to let you know that your manuscript is now accepted for publication in Life Science Alliance. Congratulations on this interesting work.

DISTRIBUTION OF MATERIALS:

Again, congratulations on a very nice paper. I hope you found the review process to be constructive and are pleased with how the manuscript was handled editorially. We look forward to future exciting submissions from your lab.

Sincerely,
